# RADIOMETRICALLY CONSISTENT GAUSSIAN SURFELS FOR INVERSE RENDERING

**Kyu Beom Han, Jaeyoon Kim, Woo Jae Kim, Jinhwan Seo & Sung-Eui Yoon**[*]
School of Computing
Korea Advanced Institute of Science and Technology
Daejeon, South Korea
`{qbhan,kimjy2630,wkim97,jinhwan.seo}@kaist.ac.kr`
`sungeui@kaist.edu`

## ABSTRACT

Inverse rendering with Gaussian Splatting has advanced rapidly, but accurately disentangling material properties from complex global illumination effects, particularly indirect illumination, remains a major challenge. Existing methods often query indirect radiance from Gaussian primitives pre-trained for novel-view synthesis. However, these pre-trained Gaussian primitives are supervised only towards limited training viewpoints, thus lack supervision for modeling indirect radiances from unobserved views. To address this issue, we introduce radiometric consistency loss, a novel physically-based constraint that provides supervision towards unobserved views by minimizing the residual between each Gaussian primitive's learned radiance and its physically-based rendered counterpart. Minimizing the residual for unobserved views establishes a self-correcting feedback loop that provides supervision from both physically-based rendering and novel-view synthesis, enabling accurate modeling of inter-reflection. We then propose Radiometrically Consistent Gaussian Surfels (RadioGS), an inverse rendering framework built upon our principle by efficiently integrating radiometric consistency by utilizing Gaussian surfels and 2D Gaussian ray tracing. We further propose a finetuning-based relighting strategy that adapts Gaussian surfel radiances to new illuminations within minutes, achieving low rendering cost (<10ms). Extensive experiments on existing inverse rendering benchmarks show that RadioGS outperforms existing Gaussian-based methods in inverse rendering, while retaining the computational efficiency.

Project Page: `https://qbhan.github.io/radiogs-page/`

## 1 INTRODUCTION

Inverse rendering, a long-standing task in computer vision and graphics, seeks to recover scene properties such as geometry, material, and illumination from one or more input images. Despite its significance, this problem remains non-trivial due to the complex interactions between light and materials, as well as the uncertainty of lighting conditions. Inspired by the remarkable success of neural radiance fields in novel view synthesis (NVS) (Mildenhall et al., 2021), recent inverse rendering techniques have adopted these implicit neural representations (Zhang et al., 2021b; Boss et al., 2021; Zhang et al., 2021a; Liang et al., 2023). More recently, Gaussian Splatting (Kerbl et al., 2023) has emerged as a powerful alternative to overcome the computational demands of implicit neural representations.

While Gaussian Splatting allows faster optimization and real-time rendering, modeling complex global illumination effects, particularly indirect illumination and inter-reflections between surfaces, remains a significant challenge. Existing Gaussian-based inverse rendering approaches often address indirect illumination as learnable residual light (Gao et al., 2024; Liu et al., 2024; Bi et al., 2024), or obtain incident radiances from NVS-trained Gaussian primitives (Liang et al., 2024; Shi et al., 2023; Sun et al., 2025; Gu et al., 2024). These approaches, however, lack supervision for

---

*Corresponding Author

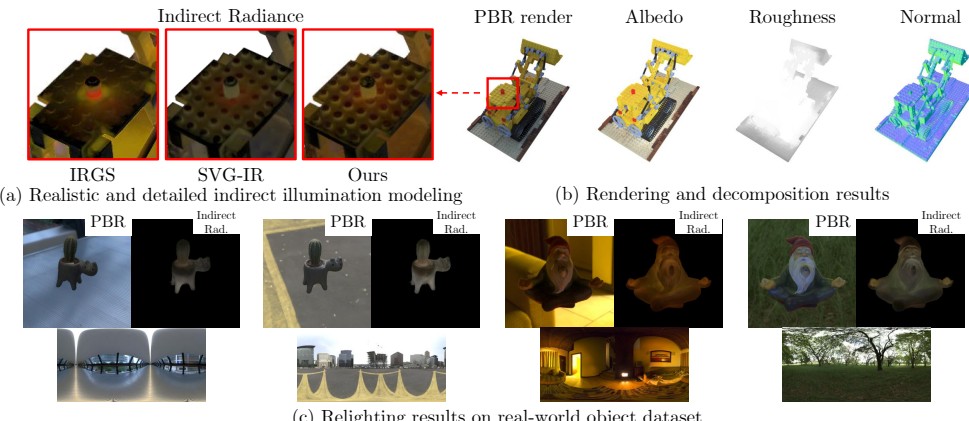

Figure 1: We introduce RadioGS, a novel inverse rendering framework that models accurate indirect illumination by providing a novel physically-based supervision on unobserved directions. (a) Compared to existing Gaussian-based methods (Gu et al., 2024; Sun et al., 2025), our method provides realistic inter-reflection between the red bulb and the blobs on the yellow lego surface, (b) leading to robust decomposition of scene properties. (c) Our method can also generate realistic indirect illumination on new lighting conditions for real objects from Stanford-ORB dataset (Kuang et al., 2023).

indirect radiances from unobserved directions due to the limited viewpoints available during NVS training. Inaccurate indirect radiances from unobserved directions may lead to incorrect surface and illumination decomposition, such as baking indirect lighting effects into the surface.

In this work, we introduce a novel physically-based constraint for Gaussian-based inverse rendering, termed radiometric consistency loss, inspired by principles from self-training neural radiance caches for global illumination (Hadadan et al., 2021; Müller et al., 2021). Radiometric consistency aims to reduce the residual between the learned radiances of Gaussian primitives and the physically-based rendered (PBR) radiances for unobserved directions, generating a self-correcting guidance between view-constrained Gaussian radiances and physical principle induced from the PBR radiances. Our physically-based constraint allows Gaussian primitives to self-correct their radiances to match the consistency and provide accurate indirect illumination for unobserved viewpoints.

We further propose RadioGS, an inverse rendering framework with efficiently integrated radiometric consistency by employing Gaussian surfels and differentiable Gaussian ray tracing. Furthermore, we introduce an efficient relighting strategy that leverages radiometric consistency to rapidly adapt Gaussian surfel radiances under novel lighting conditions, enabling per-frame rendering time below 10ms by directly utilizing adapted surfel radiances. Thorough experiments on multiple inverse rendering benchmarks demonstrate that RadioGS shows enhanced disentanglement of inter-reflections from material and geometry reconstruction, leading to superior relighting performance compared to existing Gaussian-based inverse rendering methods both quantitatively and qualitatively. In summary, our main contributions are as follows:

- Radiometric consistency, a novel physically-based constraint that guides Gaussian surfels to self-correct their radiance by enforcing consistency between surfel radiance and physically rendered radiance for unobserved viewpoints.

- RadioGS, a novel inverse rendering framework that efficiently integrates radiometric consistency based on 2D Gaussian ray tracing to accurately model indirect illumination for enhanced inverse rendering performance.

- An efficient relighting method that adapts Gaussian surfel radiances under new lighting conditions within a few minutes, achieving notable reduction in rendering time (<10ms).

## 2    RELATED WORKS

**Inverse Rendering with Neural Radiance Fields.** Inverse rendering aims to recover and decompose scene properties such as geometry, material, and lighting conditions from images (Marschner, 1998). Inspired by the success of neural radiance fields (NeRFs) (Mildenhall et al., 2021) for novel view synthesis (NVS), several works leverage NeRF-like neural representations to optimize scene properties (Zhang et al., 2021a;b; Srinivasan et al., 2021) guided by the rendering equation (Kajiya, 1986). Another line of research focuses on modeling indirect illumination to achieve an improved disentanglement of lighting conditions using NeRFs (Zhang et al., 2022; Yao et al., 2022; Zhang et al., 2023; Li et al., 2024) or directly queries indirect radiance from NVS pre-trained radiance fields (Jin et al., 2023). Subsequent works deploy path tracing (Wu et al., 2023; Dai et al., 2024) or devise enhanced sampling strategies (Attal et al., 2024) to query incident radiances for physically-based rendering with NeRFs. However, modeling indirect illumination with NeRFs is computationally intensive due to its reliance on volumetric ray marching and numerous neural network queries per ray. Our method provides an efficient representation for modeling indirect illumination with physically-based constraints for inverse rendering.

**Inverse Rendering with Gaussian Splatting.** Recent advances leverage Gaussian primitives (Kerbl et al., 2023) to encode geometry and material information, enabling fast optimization for inverse rendering (Liu et al., 2024; Bi et al., 2024). However, modeling indirect illumination with Gaussian primitives remains a key challenge. Existing methods model indirect radiances as per-Gaussian learnable parameters (Gao et al., 2024; Bi et al., 2024; Ye et al., 2025), but the unconstrained optimization may lead to ambiguous decomposition of illumination and material information. Another line of work queries indirect radiances from NVS-pretrained Gaussian primitives by baking irradiance volumes (Liang et al., 2024) or using point-based ray tracing (Sun et al., 2025). Yet, NVS-pretrained Gaussian primitives are supervised only towards observed directions, lacking supervision along arbitrary directions for indirect radiances. Recent work (Gu et al., 2024) leverages differentiable Gaussian ray tracing to optimize indirect radiances, but the training signal is still derived from synthesizing images of observed views. In contrast to these approaches, our work introduces physically-based supervision for unobserved viewpoints by enforcing all Gaussian radiances to satisfy the principle of the rendering equation.

**Self-training Radiance Caches for Global Illumination.** Efficiently evaluating the rendering equation (Kajiya, 1986) is central to both rendering and inverse rendering. Classical radiosity (Goral et al., 1984; Immel et al., 1986) solves a simplified, diffuse form of rendering equation via linear systems, while radiance caching (Krivánek et al., 2005; Krivanek & Gautron, 2022) amortizes the computational cost by storing and interpolating light samples. Recent work shows that neural caches can be self-trained to satisfy the rendering equation by iteratively minimizing the rendering-equation residual (Müller et al., 2021; Hadadan et al., 2021), and that such caches provide effective supervision on global illumination for differentiable rendering (Hadadan et al., 2023). We therefore propose an inverse rendering framework that extends these principles to Gaussian primitives, efficiently guiding Gaussian primitives to represent global illumination.

## 3    PRELIMINARIES

**Gaussian Surfels**, also termed 2D Gaussian Splatting (2DGS) (Huang et al., 2024), represent a scene with disk-like 2D Gaussian primitives, which are derived form of 3D Gaussian primitives. A Gaussian surfel is expressed using a transformation matrix $\mathbf{H} \in \mathbb{R}^{4\times 4}$ that transforms the surfel's local UV space to world space as below:

$$\mathbf{H} = \begin{bmatrix} s_u\mathbf{t}_u & s_v\mathbf{t}_v & \mathbf{0} & \mathbf{p} \\ 0 & 0 & 0 & 1 \end{bmatrix}, \tag{1}$$

where $\mathbf{t}_u$, $\mathbf{t}_v$, $\mathbf{s} = (s_u, s_v)$, and $\mathbf{p}$ refer to the two principal tangential vectors, the scaling vector, and the center position, respectively.

Ray-splat intersection is employed to determine the contribution of surfels for final rendering. A Gaussian surfel contains an opacity $\alpha$ and a view-dependent radiance attribute $c$ parameterized by learnable spherical harmonics coefficients $\text{SH}_j$. Each pixel is rendered by alpha-blending of $N$

depth-sorted Gaussian surfels:

$$\mathcal{C} = \sum_{j=1}^{N} T_j \alpha_j c_j, \ \ T_j = \prod_{k=1}^{j-1} (1 - \alpha_k), \ \ c_j = \text{SH}_j(\omega_o), \tag{2}$$

where $\mathcal{C}$ is the final pixel color, $T_j$ is the accumulated transmittance, and $\text{SH}_j$ is the spherical harmonics coefficients parameterization of $c_j$. We utilize the Gaussian surfels as the baseline for our inverse rendering framework for robust geometry recovery, and its integration with Gaussian ray tracing (described in Sec. 4.1.2).

**Physically-based Rendering** (PBR) models the interaction between light and surfaces in a scene via the rendering equation (Kajiya, 1986). The outgoing radiance $L(x, \omega_o)$ at a surface point $x$ in direction $\omega_o$ is defined as follows:

$$L(x, \omega_o) = \int_{\Omega} f_r(x, \omega_o, \omega_i) L_i(x, \omega_i)(\omega_i \cdot n_x) d\omega_i \tag{3}$$

where $f_r$ is the bidirectional reflectance distribution function (BRDF), $n_x$ is the normal at point $x$, and $L_i(x, \omega_i)$ is the incoming radiance at the point $x$ in direction $\omega_i$.

We assume the target materials for inverse rendering are mostly dielectric, where the diffuse and specular reflectance, $f_d$ and $f_s$, of a surface point $x$ are governed by diffuse albedo $a(x)$ and roughness $r(x)$, respectively. These parameters define the total reflectance $f_r$ based on a simplified Disney BRDF (Burley & Studios, 2012) to model the reflectance as below:

$$f_r(x, \omega_o, \omega_i) = f_d(x) + f_s(x, \omega_o, \omega_i) = \frac{a(x)}{\pi} + \frac{DFG}{4(n_x \cdot \omega_i)(n_x \cdot \omega_o)}, \tag{4}$$

where $D$, $F$, and $G$ are the normal distribution function, the Fresnel term, and the geometry term, respectively, which depend on roughness $r(x)$.

Incoming radiance $L_i$ may result directly from the light source or through indirect bounces of other surfaces, depending on the visibility at the surface. Thus, we model the incoming radiance as below:

$$L_i(x, \omega_i) = V(x, \omega_i) \cdot L_{dir}(x, \omega_i) + L_{ind}(x, \omega_i), \tag{5}$$

where $V$ is the visibility at the surface point $x$ with respect to the direction $\omega_i$, and $L_{dir}$ and $L_{ind}$ are the corresponding direct and indirect incident radiance terms. We note that $L_{dir}$ is independent of $x$ when light sources are distant.

## 4 METHOD

In this section, we first introduce our novel physically-based regularization termed radiometric consistency. Building on this, we present our inverse rendering framework called Radiometrically Consistent Gaussian Surfels (RadioGS), followed by our efficient relighting method based on our radiometric consistency.

### 4.1 RADIOMETRIC CONSISTENCY FOR GAUSSIAN SURFELS

Modeling accurate indirect illumination and inter-reflections between Gaussian surfels is crucial for robust decomposition of lighting and material information. Recent GS-based inverse rendering methods query indirect radiance directly from Gaussian surfels, but the surfel radiances are supervised only through reconstruction from the training images. As a result, surfel radiances along directions that are unseen by camera rays can lead to arbitrary values while still fitting the training images, degrading the stability and accuracy of indirect illumination (top-right diagram of Fig. 2). To address this issue, we introduce radiometric consistency, a novel physically-based constraint that guides Gaussian surfel radiances for unobserved directions based on the physically-based rendering process (bottom-right diagram of Fig. 2).

#### 4.1.1 FORMULATION

We consider a set of Gaussian surfels $\mathbf{G} = \{\mathcal{G}_j\}$ pretrained for novel-view synthesis (NVS) with each surfel $\mathcal{G}_j$. Let us denote the surfel radiance at position $x$ towards direction $\omega_o$, as $L_{\mathbf{G}}(x, \omega_o)$,

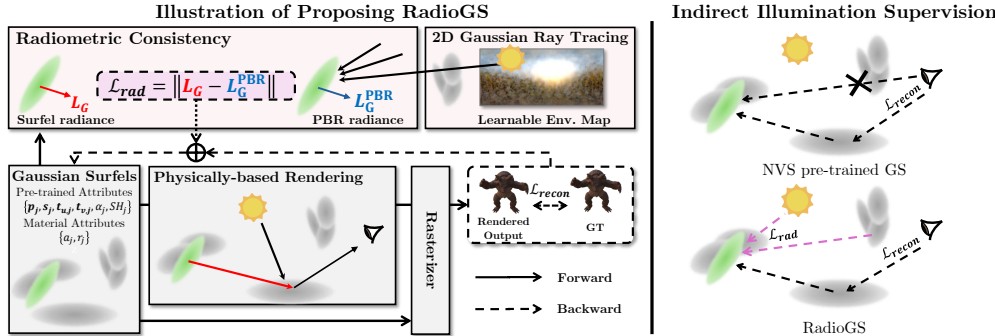

Figure 2: **Overview of our RadioGS. Left:** Our radiometric consistency loss $\mathcal{L}_{rad}$ provides physically-based supervision on indirect radiances from views unobserved by image reconstruction loss $\mathcal{L}_{recon}$, by enforcing consistency between surfel radiance $L_{\mathbf{G}}$ and physically-based rendered (PBR) radiance $L_{\mathbf{G}}^{\mathbf{PBR}}$ of Gaussian surfels. Radiometric consistency is seamlessly integrated into the inverse rendering framework, guiding Gaussian surfels to obtain physically-based radiance for delivering realistic indirect radiance to other surfels. 2D Gaussian ray tracing is deployed to jointly optimize ray-traced Gaussian surfels with our radiometric consistency loss. **Right:** Black-dotted arrows show NVS supervision, which leaves the occluded green Gaussian unconstrained. Pink-dotted arrows show our radiometric consistency providing additional supervision on its outgoing radiance along unseen directions (e.g., towards other Gaussian surfels).

which is queried from spherical harmonics coefficients of the corresponding Gaussian surfel. Each surfel has optimizable parameters for albedo and roughness. Direct illumination $L_{dir}(\omega_i)$ is represented by an environment cubemap for inverse rendering.

The core principle of our radiometric consistency is that the learned outgoing radiance of a surfel should match its physically-rendered radiance, as dictated by the rendering equation (Eq 3). We formulate our principle as a residual minimization problem. Following Eq. 3, the residual $r_{\mathbf{G}}$ can be expressed as the difference between the surfel radiance $L_{\mathbf{G}}$ and the physically-based rendered radiance $L_{\mathbf{G}}^{\mathbf{PBR}}$ as below:

$$L_{\mathbf{G}}^{\mathbf{PBR}}(x, \omega_o) = \int_\Omega f_r\left(x, \omega_o, \omega_i; \mathbf{G}\right)\left(V(x, \omega_i; \mathbf{G})L_{dir}(\omega_i) + L_{ind}(x, \omega_i; \mathbf{G})\right)(\omega_i \cdot n_x)d\omega_i, \quad (6)$$

$$\mathcal{R}_{\mathbf{G}}(x, \omega_o) = L_{\mathbf{G}}(x, \omega_o) - L_{\mathbf{G}}^{\mathbf{PBR}}(x, \omega_o), \quad (7)$$

where $L_{\mathbf{G}}^{\mathbf{PBR}}$ is the radiance calculated by physically-based rendering, $f_r(\cdot; \mathbf{G})$, $V(\cdot; \mathbf{G})$, and $L_{ind}(\cdot; \mathbf{G})$ are the BRDF, visibility, and indirect light induced by Gaussian surfels $\mathbf{G}$ based on Eq. 4 and Eq. 5, respectively.

Our radiometric consistency aims to reduce the $l_1$-norm of the residual over all Gaussian surfels and all possible directions $\omega_o$ denoted as $\mathcal{L}_{rad}$:

$$\mathcal{L}_{rad}(\mathbf{G}) = \mathbb{E}_{j,\omega_o}\left[\|\mathcal{R}_{\mathbf{G}}\|_1\right]. \quad (8)$$

Minimizing the residual norm $\|\mathcal{R}_{\mathbf{G}}\|_1$ establishes a self-correcting feedback loop based on the rendering equation. On one hand, the physically-rendered radiance $L_{\mathbf{G}}^{\mathbf{PBR}}$ serves as a physically grounded target, guiding the surfel radiance $L_{\mathbf{G}}$ to represent global illumination for unobserved viewpoints, based on the rendering equation. On the other hand, the well-constrained surfel radiances $L_{\mathbf{G}}$ towards camera viewpoints provide a strong supervisory signal that is propagated to the surfel radiances contributing to the indirect illumination term $L_{ind}$ of Eq. 6. This synergistic process allows the Gaussian surfels to obtain physically grounded radiances, thereby providing physically-induced illumination for other surfels.

### 4.1.2 2D GAUSSIAN RAY TRACING AND MONTE CARLO SAMPLING

Obtaining the visibility $V(\cdot; \mathbf{G})$ and indirect radiance $L_{ind}(\cdot; \mathbf{G})$ from Gaussian surfels is critical for creating our self-correcting feedback loop based on the inter-reflection among surfels. Point-based

ray tracing has been applied to precompute visibility (Gao et al., 2024; Guo et al., 2024) and to query indirect radiance (Sun et al., 2025) from Gaussian primitives, but lacks the differentiability and speed required for use during optimization. Inspired by recent works leveraging differentiable Gaussian ray tracing (Moenne-Loccoz et al., 2024; Xie et al., 2024), we deploy a 2D Gaussian ray tracer from IRGS (Gu et al., 2024) to leverage optimization through ray-traced surfels for radiometric consistency. 2D Gaussian ray tracing brings seamless integration with our Gaussian surfels by sharing the same ray-splat intersection that defines the contribution of Gaussian surfels.

Given a ray with the origin $x$ and the direction $\omega_i$, our ray tracer $\text{Trace}(x, \omega_i; \mathbf{G}) = (L_{trace}, T_{trace})$ gathers Gaussian surfels intersecting the ray and returns accumulated radiance $L_{trace}$ and the final transmittance $T_{trace}$ following the alpha-blending process of Eq. 2. We use ray-traced radiance $L_{trace}$ directly as indirect radiance $L_{ind}(x, \omega_i; \mathbf{G})$ and the complement of transmittance $1 - T_{trace}$ as visibility $V(x, \omega_i; \mathbf{G})$), respectively. Using our ray tracer, we acquire the Monte Carlo estimate of the integral in Eq. 7 as below:

$$L_{\mathbf{G}}^{\mathbf{PBR}}(x, \omega_o) \approx \frac{2\pi}{N_s} \sum_{i=1}^{N_s} f_r(x, \omega_o, \omega_i; \mathbf{G}) \left( V(x, \omega_i; \mathbf{G}) L_{dir}(\omega_i) + L_{ind}(x, \omega_i; \mathbf{G}) \right) (\omega_i \cdot n_x), \quad (9)$$

where we uniformly sample $N_s$ incident directions $\omega_i$ over the hemisphere defined by the surfel normal $n_x$.

We also perform Monte Carlo sampling on Gaussian surfels $\mathbf{G}$ and direction $\omega_o$ for residual estimation. We randomly sample $N_g$ surfels for each optimization step, and also sample random directions on the hemisphere defined by the normal of each sampled surfel to generate guidance towards unobserved directions. In addition, we additionally sample the directions towards camera viewpoint to propagate well-constraint supervisory signal to ray-traced Gaussian surfels. In conclusion, our design for residual estimation allows us to efficiently deploy radiometric consistency, generating self-correcting training signals explicitly for surfel radiance $L_{\mathbf{G}}$ and PBR radiance $L_{\mathbf{G}}^{\mathbf{PBR}}$ to satisfy the physical constraint of the rendering equation.

## 4.2 Inverse rendering with Radiometrically Consistent Gaussian Surfels

In this section, we introduce our inverse rendering framework RadioGS, optimizing Gaussian surfels for inverse rendering under the physically-based constraints from our radiometric consistency. Our framework operates in two stages to ensure both stable training and accuracy. We then introduce our efficient relighting strategy based on radiometric consistency.

**Initialization.** Existing works initialize geometry via NVS pre-training prior to tackling inverse rendering. To incorporate our physically-based constraint during initialization, we additionally introduce a simplified version of our radiometric consistency loss, using an efficient split-sum approximation (Munkberg et al., 2022) instead of the Monte Carlo estimate. Our approximation avoids training instability from oscillating geometry during the early optimization stage, resulting in a robust geometric foundation that is efficiently regularized based on our physically-based constraint (see the table of Figure 6 for ablation). Following 2DGS (Huang et al., 2024), we apply image reconstruction loss $\mathcal{L}_{recon}$ to images rasterized by surfel radiance $L_{\mathbf{G}}$, depth distortion loss $\mathcal{L}_{dist}$, normal-depth consistency loss $\mathcal{L}_n$, normal smoothing loss $\mathcal{L}_{ns}$, and mask-entropy loss $\mathcal{L}_{mask}$. We also add image reconstruction loss $\mathcal{L}_{recon}^{\mathbf{PBR}}$ to images rasterized by physically-rendered radiance $L_{\mathbf{G}}^{\mathbf{PBR}}$, which is approximated using the split-sum approximation. Thus, the total loss for the initialization stage is a weighted sum of the loss components as below:

$$\mathcal{L}_{init} = \mathcal{L}_{recon} + \mathcal{L}_{recon}^{\mathbf{PBR}} + \lambda_{rad}\mathcal{L}_{rad} + \lambda_{dist}\mathcal{L}_{dist} + \lambda_n\mathcal{L}_n + \lambda_{ns}\mathcal{L}_{ns} + \lambda_m\mathcal{L}_m. \quad (10)$$

**Inverse Rendering.** With our initialized Gaussian surfels, we proceed to the main inverse rendering stage by leveraging the full Monte Carlo-estimated radiometric consistency loss $\mathcal{L}_{rad}$ to accurately model complex inter-reflections. We additionally use smoothing losses for rasterized albedo and roughness, denoted as $\mathcal{L}_{as}$ and $\mathcal{L}_{rs}$, to encourage spatial coherence of material features. Finally, a light prior loss $\mathcal{L}_{light}$ (Liu et al., 2023) is applied to encourage the rendered incident diffuse illumination to adopt a natural white appearance. Thus, the total optimization objective for inverse rendering is a weighted sum of loss components as below:

$$\mathcal{L}_{inv} = \mathcal{L}_{init} + \lambda_{as}\mathcal{L}_{as} + \lambda_{rs}\mathcal{L}_{rs} + \lambda_{light}\mathcal{L}_{light}. \quad (11)$$

Table 1: **Quantitative comparisons on TensoIR dataset** (Jin et al., 2023). The results are colored in rank as 1st, 2nd, and 3rd. Our method surpasses existing Gaussian-based methods and a NeRF-based method in most metrics, while maintaining the computational efficiency with the average training time of 1 hour. We report our relighting metric using Gaussian ray tracing (Ours) and finetuning-based method (Ours*).

| Method | Novel View Synthesis | | | Normal | Albedo | | | Relight | | | Training |
|---|---|---|---|---|---|---|---|---|---|---|---|
| | PSNR ↑ | SSIM ↑ | LPIPS ↓ | MAE ↓ | PSNR ↑ | SSIM ↑ | LPIPS ↓ | PSNR ↑ | SSIM ↑ | LPIPS ↓ | hours |
| TensoIR | 35.09 | 0.976 | 0.040 | 4.100 | 29.27 | 0.950 | 0.085 | 28.58 | 0.944 | 0.081 | 4 |
| GS-IR | 35.33 | 0.974 | 0.039 | 4.948 | 29.94 | 0.921 | 0.100 | 24.37 | 0.885 | 0.096 | 0.5 |
| GI-GS | 36.75 | 0.972 | 0.037 | 5.253 | 29.90 | 0.921 | 0.099 | 24.70 | 0.886 | 0.106 | 0.5 |
| R3DG | 33.35 | 0.964 | 0.041 | 5.927 | 29.27 | 0.951 | 0.078 | 27.37 | 0.909 | 0.083 | 1.1 |
| IRGS | 35.43 | 0.964 | 0.049 | 4.209 | 30.62 | 0.956 | 0.072 | 29.91 | 0.935 | 0.076 | 0.9 |
| SVG-IR | 36.71 | 0.976 | 0.033 | 4.358 | 30.48 | 0.950 | 0.074 | 31.10 | 0.946 | 0.056 | 1.1 |
| Ours | 37.86 | 0.980 | 0.027 | 3.689 | 31.05 | 0.952 | 0.072 | 32.09 | 0.953 | 0.048 | 1.0 |
| Ours* | | | | | | | | 31.41 | 0.948 | 0.052 | |

Please refer to A for the details of additional loss functions $\mathcal{L}_{recon}$, $\mathcal{L}_{recon}^{\mathbf{PBR}}$, $\mathcal{L}_{ns}$, $\mathcal{L}_m$, $\mathcal{L}_{as}$, $\mathcal{L}_{rs}$, and $\mathcal{L}_{light}$, and the learning rates of Gaussian surfel parameters.

**Relighting.** Once the lighting condition changes, surfel radiances cannot provide indirect illumination, since they are specifically optimized for the previous lighting condition. Instead, we query indirect radiances following IRGS (Gu et al., 2024) by alpha-blending the normal, albedo, and roughness towards the incident direction using Gaussian ray tracing, and applying a split-sum approximation to efficiently estimate the incident radiance. However, storing numerous incident radiances per surfel and re-estimating outgoing radiances based on Eq. 9 consumes additional rendering time.

To this end, we introduce a finetuning-based relighting approach that leverages radiometric consistency. Radiometric consistency allows surfel radiances to rapidly adapt to new lighting conditions. Given a new lighting condition, we perform a few finetuning iterations exclusively on the surfel radiances by minimizing our radiometric consistency loss $\mathcal{L}_{rad}$. Once finetuning is complete, the scene can be rendered from any viewpoint using only surfel radiances. Please refer to Figure 19 of Appendix.J for additional illustration on our finetuning method.

## 5 EXPERIMENTS

### 5.1 EXPERIMENTAL SETUPS

**Dataset and Metric.** We evaluate our method's novel view synthesis (NVS), inverse rendering, and relighting capabilities using two synthetic datasets: TensoIR (Jin et al., 2023) and Synthetic4Relight (Zhang et al., 2022). These two synthetic datasets provide diverse lighting conditions and ground truth for geometry and material evaluation. We employ PSNR, SSIM, and LPIPS for evaluating NVS, albedo, and relighting. Normal reconstruction is evaluated using Mean Angular Error (MAE), and roughness is evaluated using Mean Square Error (MSE). We also provide qualitative relighting results on a real-world object dataset Stanford-ORB (Kuang et al., 2023) in Figure 1-(c).

**Implementation Details.** For our radiometric consistency loss $\mathcal{L}_{rad}$, we set the weight $\lambda_{rad} = 0.2$. We sample $N_g = 4096$ Gaussian surfels and $N_s = 64$ incident rays per surfel, resulting in $2^{18}$ rays traced through Gaussian surfels to calculate the radiometric consistency loss at every training iteration. We store the ray-traced results on sampled Gaussians at each step for use in physically-rendered image. For our relighting method, we set the weight $\lambda_{rad} = 1.0$, and discard all other losses. Experiments were conducted on an NVIDIA RTX 4090 GPU, with total optimization taking approximately 60 minutes (30 for initialization and 30 for inverse rendering), and the finetuning process taking approximately 2 minutes. Please refer to Appendix.A.4 for further details.

**Baselines.** We compare our method against prior Gaussian Splatting (GS)-based methods: GS-IR (Liang et al., 2024), GI-GS (Chen et al., 2024), R3DG (Gao et al., 2024), IRGS (Gu et al., 2024), and SVG-IR (Sun et al., 2025). We also include TensoIR (Jin et al., 2023), an efficient NeRF-based approach. Quantitative and qualitative results are reproduced using the publicly available code.

### 5.2 INVERSE RENDERING PERFORMANCE COMPARISONS

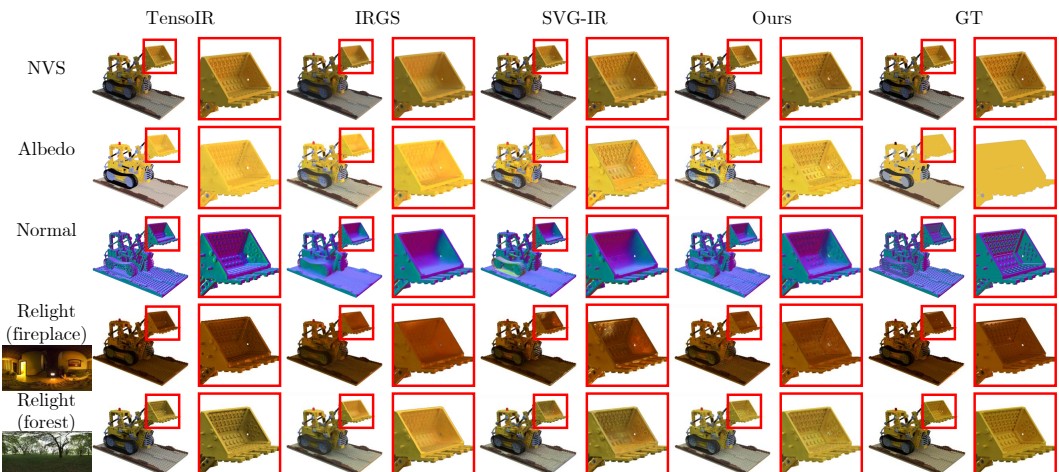

Figure 3: **Qualitative result on the "lego" scene of TensoIR dataset**. Our method provides enhanced decomposition and realistic relighting results compared to Gaussian-based methods. Specifically, our method shows noticeably robust performance on regions with high geometric complexity, such as the highlighted bucket. Best viewed in zoom.

**TensoIR.** On the TensoIR dataset (Table 1), our approach demonstrates superior performance on various metrics including novel-view synthesis (NVS), normal estimation, and relighting compared to existing methods. Notably, our method outperforms other ray-tracing based methods on Gaussian primitives (Gu et al., 2024; Sun et al., 2025), reflecting the necessity of physically-based constraints on surfel radiances in inverse rendering. Moreover, our finetuning-based relighting method outperforms existing relighting methods, indicating the effectiveness of our self-correcting guidance from radiometric consistency.

Qualitative results on Figure 3 illustrate our method's performance on reconstructing finer geometric details for normal reconstruction and NVS, which leads to more realistic relighting results. Figure 1-(a) showcases the realistic and detailed indirect illumination modeled by our method on the same scene compared to the other competitors.



Figure 4: **Relighting results on the "armadillo" scene of TensoIR dataset.**

Additional comparisons on relighting (Figure 4) show that both of our relighting methods achieve realistic relighting results, showing real-time rendering capabilities. Especially, our finetuning-based method shows the fastest rendering time, with a minor compromise in quality compared to ray-tracing based relighting. This is because finetuning process accumulates minor errors from estimated geometry and material properties of Gaussian surfels into surfel radiances, leading to the trade-off in visual quality. Figure 20 of Appendix.J contains additional comparison of our relighting method.

**Synthetic4Relight.** Results on the Synthetic4Relight dataset (Table 2) further validate the capabilities of our method, outperforming existing methods in NVS, albedo reconstruction and relighting, while showing comparable performance on roughness estimation. Visual comparisons on Figure 5 demonstrate how our realistic modeling of indirect illumination leads to enhanced albedo reconstruction and NVS. The inter-reflecting directions of the highlighted region are

Table 2: **Quantitative comparisons on Synthetic4Relight dataset.**

| Method | NVS PSNR ↑ | Roughness MSE ↓ | Albedo PSNR ↑ | Relight PSNR ↑ |
|--------|-----------|-----------------|---------------|----------------|
| R3DG | 34.10 | 0.010 | 28.65 | 33.12 |
| IRGS | 34.44 | **0.008** | 30.50 | 34.35 |
| SVG-IR | 34.14 | 0.009 | 29.06 | 32.59 |
| Ours | **34.98** | 0.011 | **30.69** | **34.87** |

overlooked during novel-view synthesis training, whereas our radiometric consistency provides physically-based constraint on surfel radiances towards reflecting directions, resulting in realistic indirect illumination. Please refer to Appendix. D for additional qualitative results.

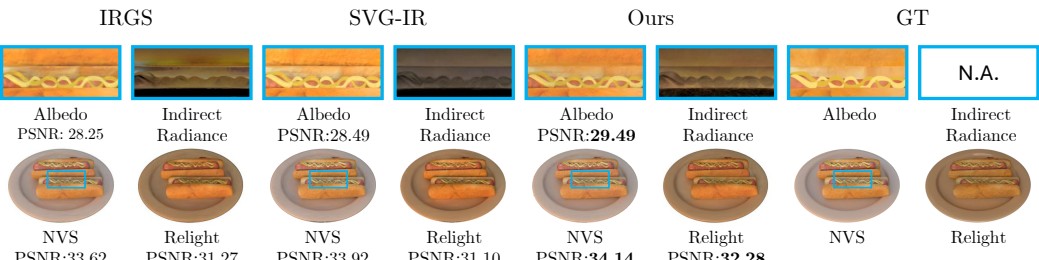

Figure 5: **Qualitative results on the "hotdog" scene of Synthetic4Relight (Zhang et al., 2022) dataset.** Our method models natural inter-reflection between the sausages and the buns, showing superior reconstruction performance on highlighted regions. IRGS shows relatively bright and fluctuating indirect illumination, which led to darker albedo reconstruction. SVG-IR models relatively darker indirect illumination, returning brighter albedo reconstruction. Best viewed in zoom.

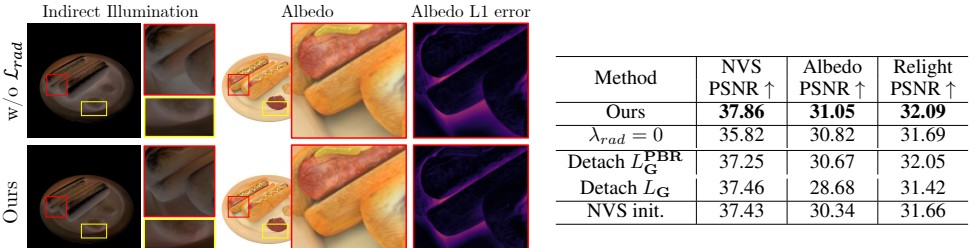

| Method | NVS PSNR ↑ | Albedo PSNR ↑ | Relight PSNR ↑ |
|---|---|---|---|
| Ours | **37.86** | **31.05** | **32.09** |
| $\lambda_{rad} = 0$ | 35.82 | 30.82 | 31.69 |
| Detach $L_{\mathbf{G}}^{\mathbf{PBR}}$ | 37.25 | 30.67 | 32.05 |
| Detach $L_{\mathbf{G}}$ | 37.46 | 28.68 | 31.42 |
| NVS init. | 37.43 | 30.34 | 31.66 |

Figure 6: **Ablation studies on our radiometric consistency.** The left sub-figure demonstrates how our radiometric consistency loss $\mathcal{L}_{rad}$ provides guidance on radiances towards unobserved views such as the interstices, leading to enhanced albedo reconstruction (red box). Also, our method guides the generation of inter-reflections between the ketchup and the plate (yellow box). The right table contains PSNR metrics for the ablation studies.

## 5.3 ABLATION STUDIES ON RADIOMETRIC CONSISTENCY

We report ablation studies on components of our radiometric consistency on the TensoIR dataset. The table of Figure 6 shows the PSNR metrics for three categories, NVS, albedo reconstruction, and relighting, in our ablation studies.

**Absence of Radiometric Consistency.** We perform ablation studies on the radiometric consistency by removing the radiometric consistency loss during the inverse rendering stage (see the left sub-figure of Figure 6 and "$\lambda_{rad} = 0$" on the right table of Figure 6). The absence of radiometric consistency provides incorrect indirect radiances on unobserved directions, degrading the albedo reconstruction on the corresponding regions and leading to significant performance degradation in all three categories. Further analysis on indirect illumination modeling and inverse rendering are provided in Appendix.E.

**Detaching Gradient Flows from $\mathcal{L}_{rad}$.** We ablate on the self-correcting gradient flow by detaching the gradients towards Gaussian surfels during the calculation of the surfel radiance $L_{\mathbf{G}}$ and physically-based rendered radiance $L_{\mathbf{G}}^{\mathbf{PBR}}$ on Eq. 7. Detaching either gradient leads to an overall performance drop. Detaching gradient from $L_{\mathbf{G}}$ cause noticeable degradation on albedo reconstruction, while detaching gradient from $L_{\mathbf{G}}^{\mathbf{PBR}}$ degrades NVS. Such degradation reflects the importance of the view-constrained supervision signal from $L_{\mathbf{G}}$, and the physically-based constraint delivered by $L_{\mathbf{G}}^{\mathbf{PBR}}$.

**Initialization.** We found that removing radiometric consistency during initialization degrades overall performance, highlighting the contribution of our radiometric consistency as beneficial physically-based guidance for initialization.

# 6 CONCLUSION AND FUTURE WORKS

We introduced a novel physically-based supervision called radiometric consistency, which addresses the key challenge of modeling indirect illumination in Gaussian-based representations by guiding Gaussian surfels to learn accurate indirect illumination towards unobserved directions. We then introduced Radiometrically Consistent Gaussian Surfels (RadioGS), a novel inverse rendering framework that efficiently leverages radiometric consistency by utilizing 2D Gaussian ray tracing. We also presented a new relighting method that leverages our constraint to quickly adapt surfel radiances to new lighting environments, achieving a rendering time below 10ms per frame. Experiments demonstrated that RadioGS outperforms existing Gaussian-based methods on two synthetic benchmarks, based on accurate and realistic indirect illumination. Since the current method only supports dielectric materials, extending radiometric consistency to more complex materials, such as anisotropic or highly-reflective surfaces, would be an interesting future direction.

## ACKNOWLEDGEMENTS

This work was supported by the Institute of Information communications Technology Planning Evaluation (IITP) grant (No. RS-2025-25443318, Physically-grounded Intelligence: A Dual Competency Approach to Embodied AGI through Constructing and Reasoning in the Real World; LG AI STAR Talent Development Program for Leading Large-Scale Generative AI Models in the Physical AI Domain (No. RS-2025-25442149), and IITP(Institute of Information & Coummunications Technology Planning & Evaluation)-ITRC(Information Technology Research Center) grant funded by the Korea government(Ministry of Science and ICT)(IITP-2026-RS-2020-II201460).

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

APPENDIX

THE USE OF LLMS

The author(s) used ChatGPT for minor grammatical adjustments, and all resulting edits were carefully reviewed and finalized by the author(s).

## A    IMPLEMENTATION DETAILS

In this section, we discuss additional details of the implementation of our work.

### A.1    DEPTH INTERPOLATION

Depth maps of our Gaussian surfels are rendered by interpolation of the Gaussian primitives:

$$\mathcal{D} = \sum_{i=1}^{N} \frac{\alpha_i T_i}{\sum_{j=1}^{N} \alpha_j T_j} d_i, \tag{12}$$

where $d_i$ is the depth of $i$-th primitive. This formulation ensures the depth map reflects the visibility-weighted contribution of all overlapping primitives.

### A.2    LOSS FUNCTIONS

#### A.2.1    RECONSTRUCTION LOSS

The reconstruction loss is composed of a weighted sum of $l_1$-loss and SSIM (WangZhou et al., 2004). Following our baseline (Huang et al., 2024), we assign a weight of 0.8 to the $l_1$-loss and 0.2 to SSIM.

#### A.2.2    DISTORTION LOSS

The depth distortion loss enforces geometric consistency along rays by minimizing the weighted pairwise depth differences between Gaussian primitives:

$$\mathcal{L}_{dist} = \sum_{i,j} \alpha_i T_i \alpha_j T_j \, |z_i - z_j|, \tag{13}$$

where $z_i$ denotes the depth value of the $i$-th primitive. The loss drives Gaussian primitives to collapse into tight clusters aligned with surface geometry and enhances depth coherence.

#### A.2.3    NORMAL-DEPTH CONSISTENCY LOSS

This loss enforces geometric coherence by aligning Gaussian primitive normals with surface geometry derived from depth gradients:

$$\mathcal{L}_n = \sum_i \alpha_i T_i (1 - n_i^T N) \tag{14}$$

where $n_i$ is the normal vector of $i$-th primitive and $N$ is the surface normal at the median of intersection $p_s$ estimated from gradient of depth map:

$$N = \frac{\nabla_x p_s \times \nabla_y p_s}{\|\nabla_x p_s \times \nabla_y p_s\|}. \tag{15}$$

#### A.2.4    FIRST-ORDER EDGE AWARE SMOOTHING LOSS

We use edge-aware smoothing constraints to enhance spatial coherence while preserving structural edges for surface normal, albedo, and roughness predictions. These losses minimize the gradient of each feature and relax smoothing constraints at image edges:

$$\mathcal{L}_{\{n,a,r\}s} = \|\nabla\{\mathcal{N}, \mathcal{A}, \mathcal{R}\}\| \exp(-\|\nabla\mathcal{C}_{gt}\|), \tag{16}$$

where $\mathcal{N}$, $\mathcal{A}$, and $\mathcal{R}$ are rendered normal, albedo, and roughness map, respectively and $\mathcal{C}_{gt}$ is the ground truth training image.

### A.2.5 Sparsity Loss

The sparsity loss drives Gaussian's opacity towards 0 or 1:

$$\mathcal{L}_s = \frac{1}{|\alpha|} \sum_{\alpha_i} [\log(\alpha_i) + \log(1 - \alpha_i)] \tag{17}$$

It collapses the spatial distribution of Gaussian primitives into thin surface-aligned layers and accelerates ray tracing by reducing hits and sorting via early termination.

### A.2.6 Light Prior Loss

The light prior loss enforces neutral white illumination in diffuse rendering. It constrains the per-channel average intensities $\bar{c}_i$ of estimated lighting:

$$\mathcal{L}_{light} = \frac{1}{3} \sum_{i=1} 3 \left| \bar{c}_i - \frac{1}{3} \sum_{j=1} 3\bar{c}_j \right| \tag{18}$$

### A.3 2D Gaussian Ray Tracer

We implemented 2D Gaussian ray tracer using Pytorch CUDA extensions and OptiX (Parker et al., 2010) following Moenne-Loccoz et al. (2024) and Gu et al. (2024). We adopt a simpler BVH construction with two triangles encapsulating the 2D Gaussian primitives from Xie et al. (2024), reducing the BVH update on each training iteration from 3ms to 2ms. The Gaussian response is achieved by analytically calculating the intersection point $p$ between the flat 2D Gaussian primitive with the center $\mu$ and normal $n$ and the ray with origin $o$ and direction $d$ as below:

$$p = \left( \frac{n \cdot (\mu - d)}{n \cdot d} \right) d + o. \tag{19}$$

Such formulation is identical to the one that of the 2DGS Huang et al. (2024) rasterizer, ensuring consistent Gaussian response between the rasterizer and the ray tracer.

To reduce the computation of depth-sorting ray-traced Gaussians, we utilize any-hit program to gather $k$ Gaussians within the buffer. Once the buffer is full, we sort the gathered Gaussians by depth, and accumulate the radiance and trasmittance based on Eq. 2. The process repeats to gather the next $k$ Gaussians until all ray-traced Gaussians are accumulated or the transmittance reaches the threshold. We use buffer of K=16 for sorting Gaussians per ray, and terminate the tracing when with the transmittance threshold of 0.03. For differentiability, we re-cast the rays to gather the same set of Gaussians, and analytically calculate the gradients.

### A.4 Additional Training Details

We use learning rates of 0.005, 0.005, 0.01 for albedo, roughness, and cubemap, respectively, and other hyperparameters following the configuration of 2DGS (Huang et al., 2024). We represent the optimizable environment map as cubemap with a resolution of 32. The first initailize stage is trained for 40K iterations, with loss weight hyperparameters $\lambda_d$, $\lambda_n$, $\lambda_{ns}$, $\lambda_s$ as 1000, 0.05, 0.02, and 0.05, respectively. The inverse rendering stage is trained for 20K iterations, with loss weight hyperparameters $\lambda_{as}$, $\lambda_{rs}$, $\lambda_{light}$ as 0.2, 0.1, and 0.01, respectively. After the initialization stage, we reinitialize the albedo, roughness, and cubemap. Then, we start the inverse rendering stage with the same learning rate depicted above. For the finetuning stage, we set the same learning rate only for the spherical harmonics coefficients.

### A.5 Rendering and Relighting with Split-sum Approximation

Split-sum approximation is a technique for efficiently computing indirect illumination in physically based rendering. By decomposing the complex specular BRDF integral into two separable terms on Eq. 3, it avoids the computational burden of Monte Carlo sampling while preserving visual fidelity.

We divide the light transport into diffuse $L_d$ and specular $L_s$ components each and approximate the specular light transport as below:

$$L_s(\omega_o) \approx \int_\Omega f_s(\omega_i, \omega_o)(\omega_i \cdot N)d\omega_i \cdot \int_\Omega L_i(\omega_i)D(\omega_i, \omega_o)(\omega_i \cdot N)d\omega_i. \tag{20}$$

This precomputation allows the specular contribution to be efficiently estimated at runtime by sampling the pre-filtered environment map (using the reflection vector and roughness) and the BRDF LUT.

Diffuse radiance $L_d$ is computed more directly as the product of the surface's diffuse reflectance (albedo) and the total incoming diffuse light. The latter is also precomputed by convolving the environment map with a cosine lobe to create an irradiance map.

We apply the split-sum approximation for the initialization stage to easily approximate the estimate of physically-rendered outgoing radiance $L_{pbr}$ on each Gaussian primitive. For relighting, we apply the split-sum approximation to calculate the incident indirect illumination $L_{ind}$ from the traced secondary ray using the ray-traced surface position, normal, albedo, and roughness values. The achieved $L_{ind}$ is used for relighting integrated with the traced visibility $V$ and the queried direct light $L_{dir}$.

## B    COMAPARISON ON RELIGHTING PERFORMANCE AND RENDERING COST

On table 3, we report relighting performance using three configurations: (1) Gaussian ray tracing that estimates indirect radiance using the PBR split-sum approximation (PBR split-sum), (2) Gaussian ray tracing that uses indirect radiance predicted by fine-tuned surfels (PBR fine-tuned), and (3) direct rasterization with fine-tuned surfel radiances (Surfel fine-tuned). While fine-tuning introduces a slight quality drop, it enables the surfel radiances to adapt to new lighting conditions and act as physically consistent indirect illumination sources, all while achieving substantially faster rendering speeds than competing approaches. We also provide qualitative comparisons between the three configurations in Figure 20.

Table 3: Relighting Performance and Rendering cost during relighting on TensoIR dataset.

| Method | Relight | | | Rendering |
|---|---|---|---|---|
| | PSNR ↑ | SSIM ↑ | LPIPS ↓ | ms |
| IRGS | 29.91 | 0.935 | 0.076 | 1090 |
| SVG-IR | 31.10 | 0.946 | 0.056 | 82.48 |
| PBR split-sum | 32.09 | 0.953 | 0.048 | 38.64 |
| PBR finetuned | 31.59 | 0.952 | 0.049 | 38.29 |
| Surfel finetuned | 31.41 | 0.948 | 0.052 | 5.902 |

## C    VISUAL COMPARISON ON ILLUMINATION COMPONENTS

We provide additional visual comparisons on "hotdog" and the "lego" scene from the TensoIR dataset (Jin et al., 2023) in Figure 7 and Figure 8. We visualize illumination components including incident direct and indirect radiances, and their rendered results on the datasets along with Gaussian-based methods IRGS (Gu et al., 2024) and SVG-IR (Sun et al., 2025) to compare our realistic indirect illumination. The components are the mean value of the samples during the Monte Carlo rendering. Our method provides realistic indirect illumination that maintains the fine details of inter-reflecting surfaces, while the IRGS (Gu et al., 2024) often overestimates the intensity of the indirect radiance and SVG-IR (Sun et al., 2025) often underestimates the intensity of indirect radiance due to the lack of physical guidance for indirect radiances on unobserved views.

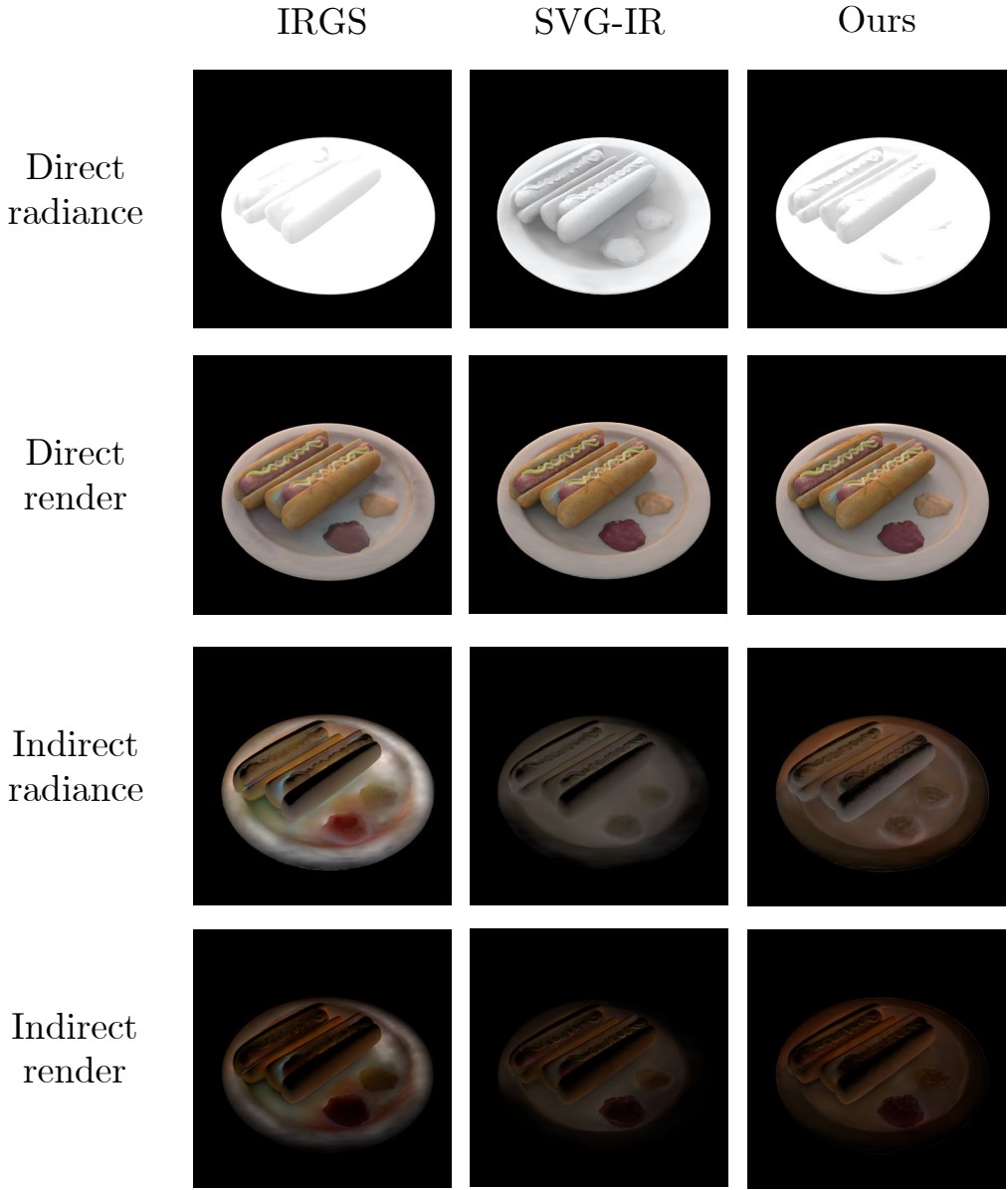

Figure 7: Qualitative comparison on illumination components on the "hotdog" scene of TensoIR dataset. Best viewed in zoom.

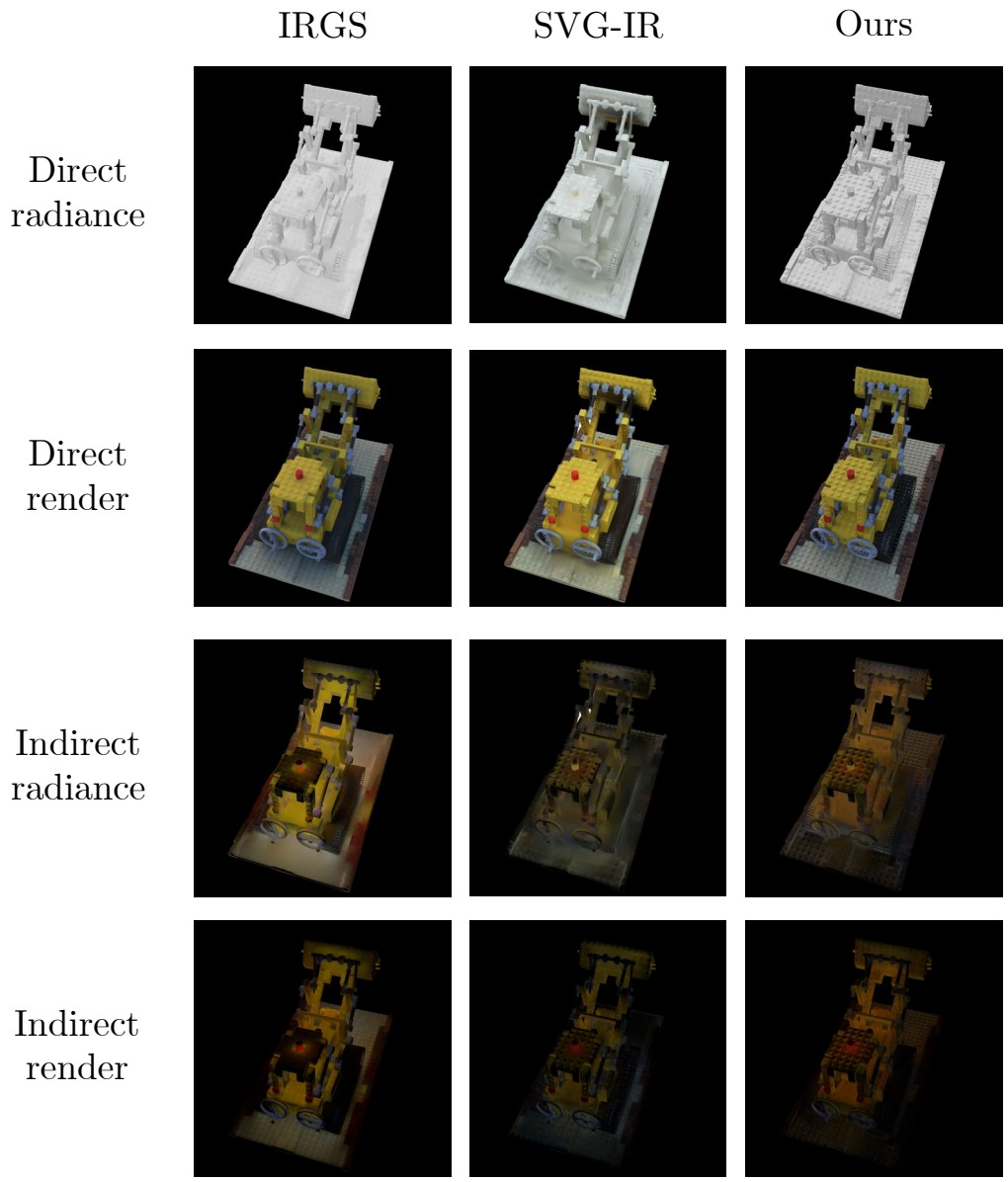

Figure 8: Qualitative comparison on illumination components on the"lego" scene of TensoIR dataset. Best viewed in zoom.

# D   ADDITIONAL VISUAL COMPARISON ON BENCHMARK DATASETS

We provide additional visual comparisons on "armadillo" scene from the TensoIR (Jin et al., 2023) dataset, and all the scenes from the Synthetic4Relight dataset from Figure 9 to 12. For the TensoIR dataset, we deliver comparison on novel-view synthesis (NVS), normal reconstruction, albedo reconstruction and relighting with Gaussian-based methods IRGS (Gu et al., 2024) and SVG-IR (Sun et al., 2025), and NeRF-based method TensoIR (Jin et al., 2023). For the Synthetic4Relight dataset, we deliver comparison on novel-view synthesis (NVS), albedo reconstruction, roughness reconstruction, and relighting with Gaussian-based methods R3DG (Gao et al., 2024), IRGS (Gu et al., 2024) and SVG-IR (Sun et al., 2025).

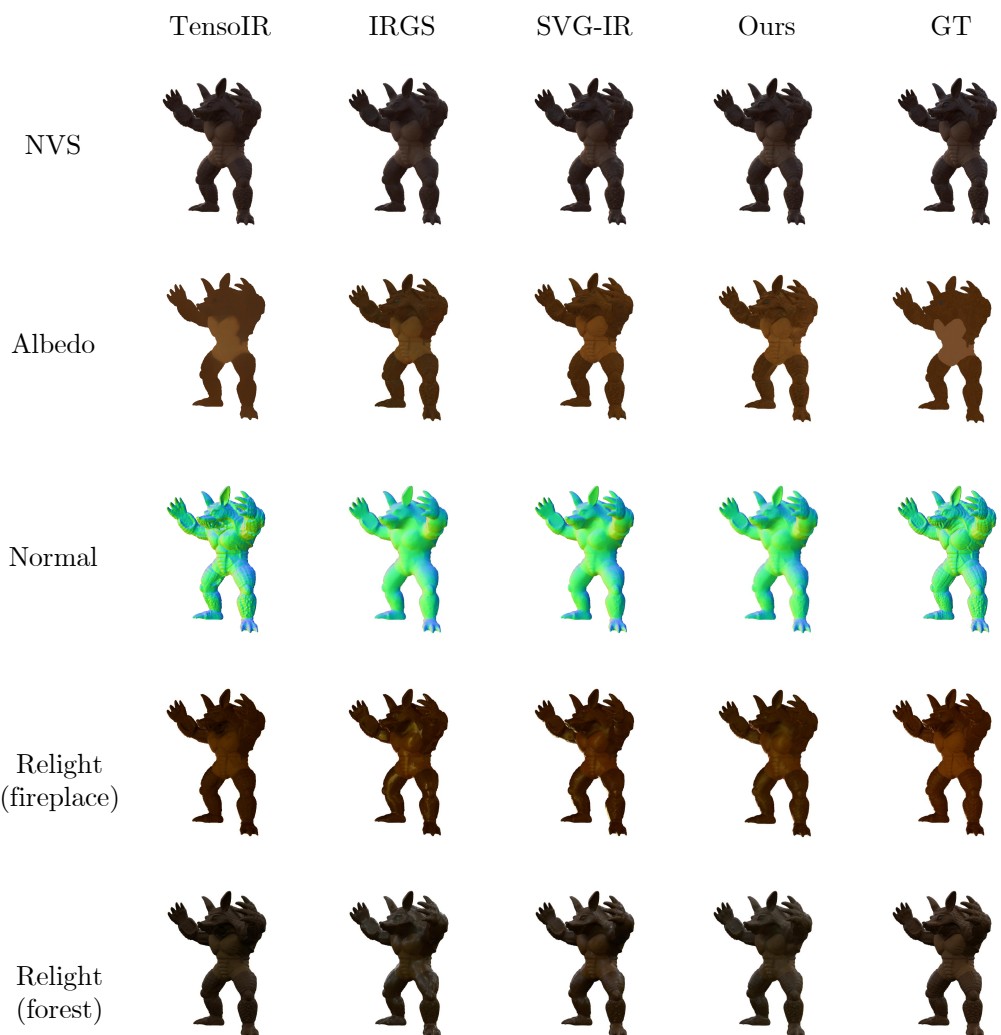

Figure 9: Qualitative comparison on NVS, albedo reconstruction, normal reconstruction, and relighting on the "armadillo" scene of the TensoIR dataset. Best viewed in zoom.

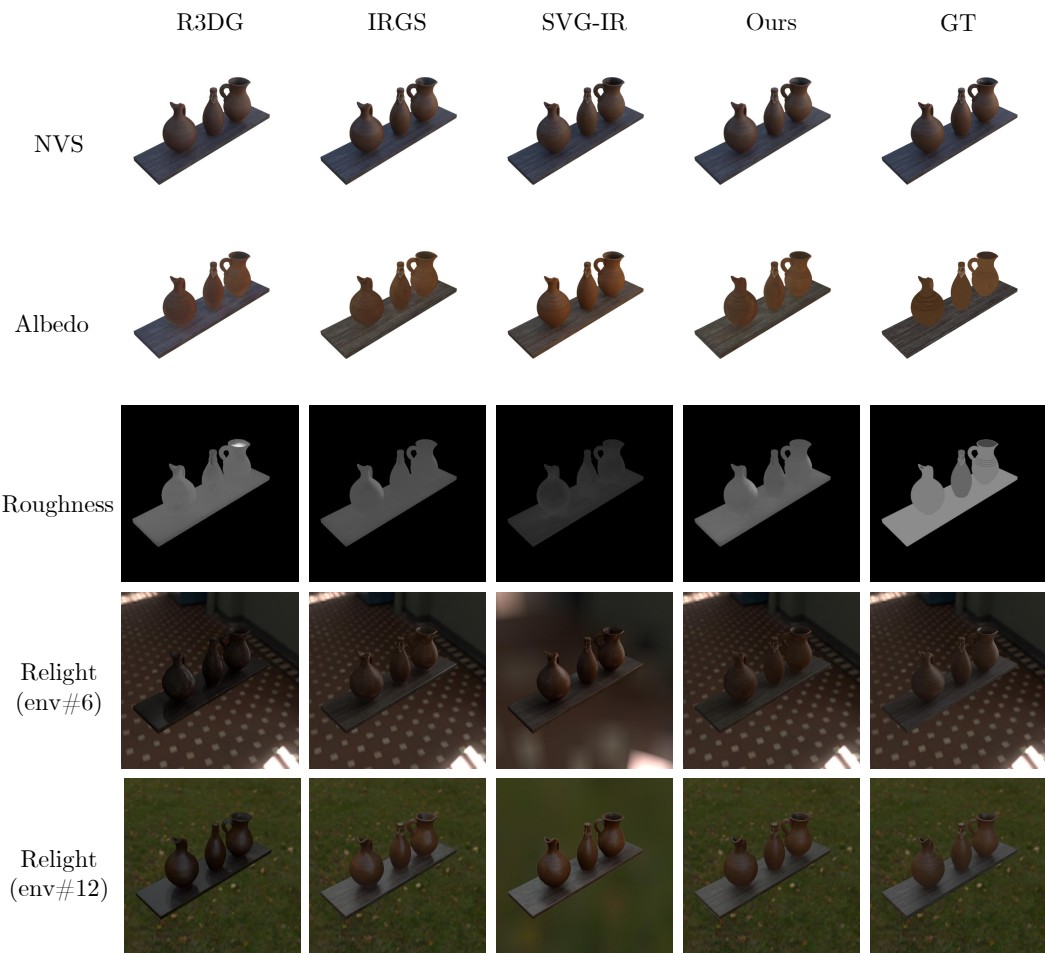

Figure 10: Qualitative comparison on NVS, albedo reconstruction, roughness reconstruction, and relighting on the "jugs" scene of the Synthetic4Relight dataset. Best viewed in zoom.

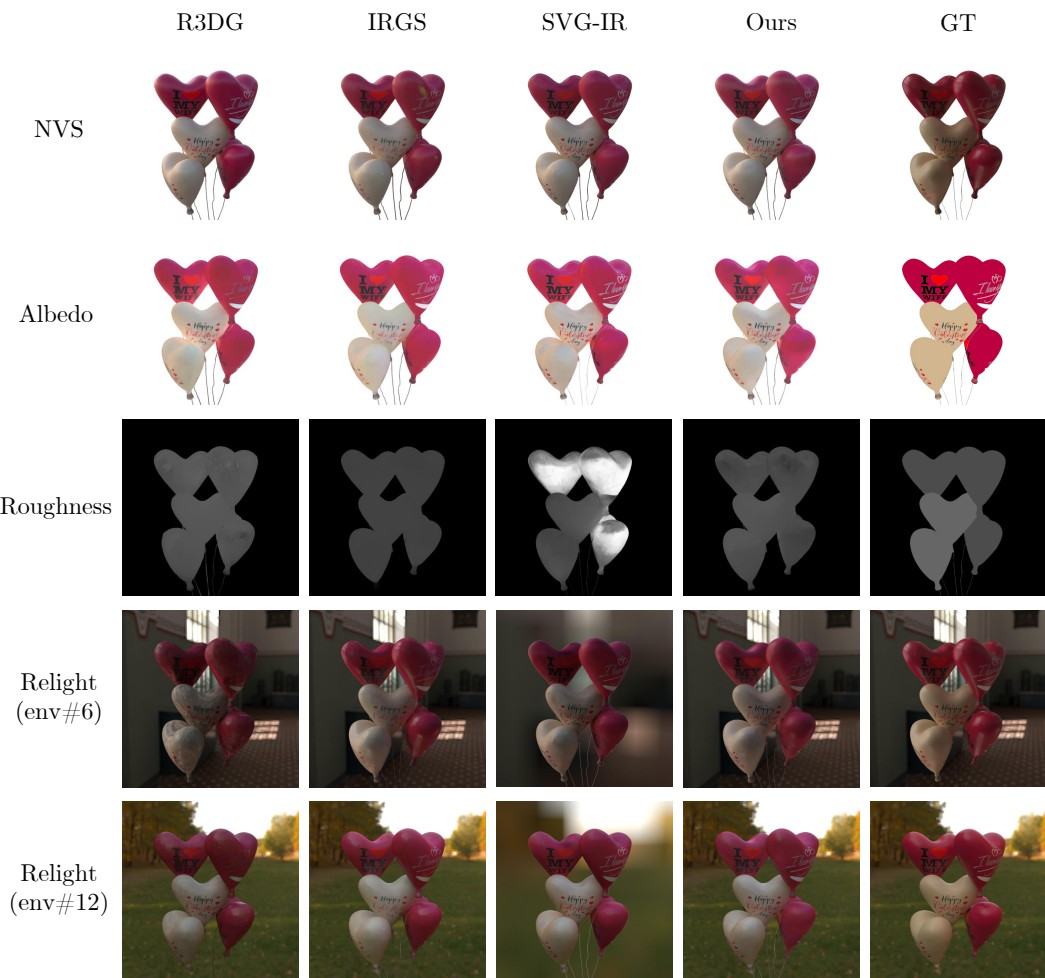

Figure 11: Qualitative comparison on NVS, albedo reconstruction, roughness reconstruction, and relighting on the "air baloons" scene of the Synthetic4Relight dataset. Best viewed in zoom.

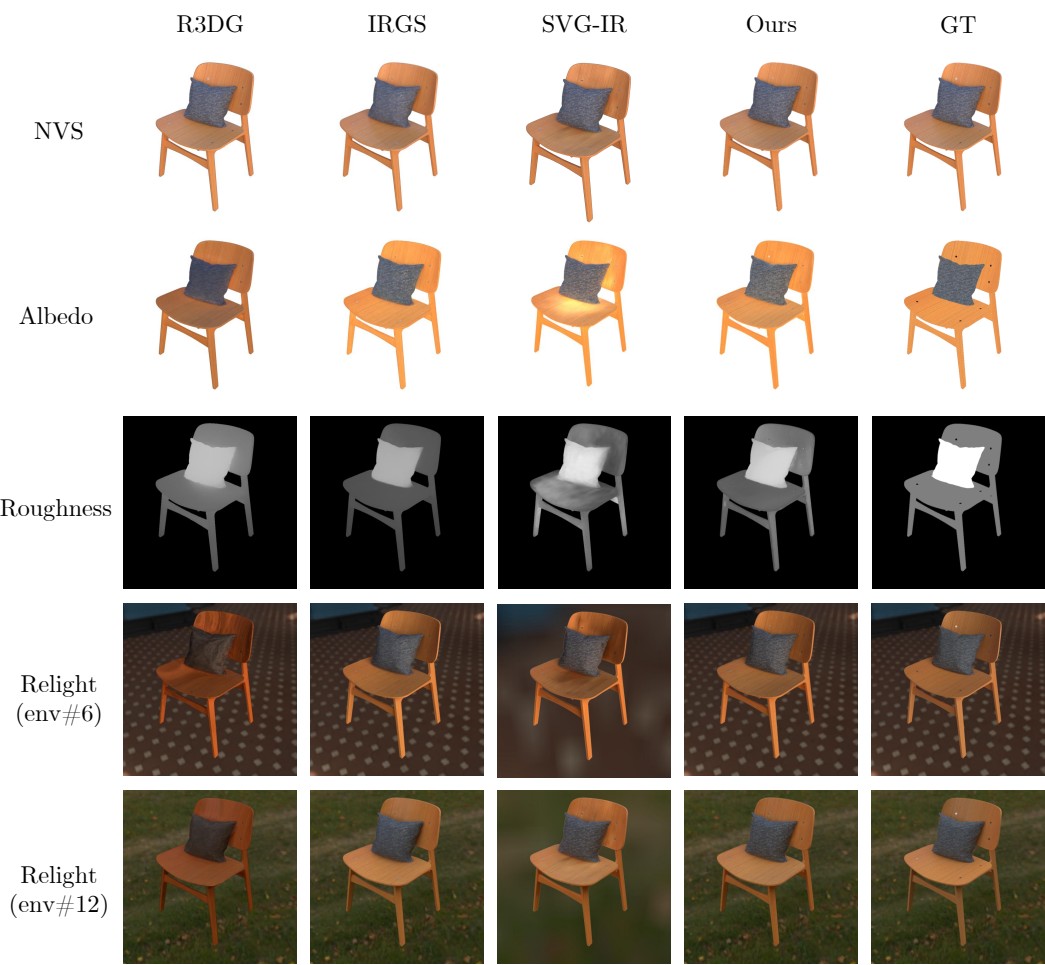

Figure 12: Qualitative comparison on NVS, albedo reconstruction, roughness reconstruction, and relighting on the "chair" scene of the Synthetic4Relight dataset. Best viewed in zoom.

# E    EVALUATION OF INDIRECT ILLUMINATION

Current inverse rendering benchmarks, including TensoIR Jin et al. (2023), Synthetic4Relight Zhang et al. (2022), and Stanford-ORB Kuang et al. (2023), do not provide ground truth (GT) for indirect illumination for quantitative and qualitative evaluation. To address this limitation, we generated a new evaluation dataset with explicit GT Indirect Illumination. We utilized the original Blender files from the TensoIR dataset to generate high-fidelity ground truth. We used Blender Cycles path tracing engine to render the indirect illumination pass along with the original render pass. To ensure noise-free references, especially for indirect illumination, we set the sampling rate to 256 spp and applied the OIDN denoiser [3]. This allows direct quantitative evaluation of the indirect illumination components. We trained our model and the ablation model discarding the radiometric consistency loss (Ours w/o $\mathcal{L}_{rad}$) using the same hyperparameters described in the paper. We also trained two baselines, IRGS and SVG-IR, for comparison.

Table 4 presents the quantitative comparison against baselines (IRGS, SVG-IR) and our ablation model on our new dataset. Our method significantly outperforms all baselines in indirect illumination reconstruction, confirming that our method accurately models the physical transport of indirect light. Our accurate indirect illumination leads to superior performance in most other metrics. When radiometric consistency is removed (Ours w/o $\mathcal{L}_{rad}$), the indirect PSNR drops significantly, showing that the performance gain comes from our proposed framework utilizing radiometric consistency, which effectively supervises indirect radiance from unobserved views.

We also provide qualitative comparisons in Figure 13. As shown, our method faithfully reconstructs indirect illumination compared to baseline models and our ablation model. Overall, our method produces indirect illumination closest to the ground truth, while IRGS produces overestimated intensity, and SVG-IR tends to underestimate the intensity of the indirect radiances. Similar phenomena can also be observed in the qualitative results in Figures 7 and 8 of our paper. Our ablation model (Ours w/o $\mathcal{L}_{rad}$) tends to produce white blurs on inter-reflecting regions compared to our method due to the lack of supervision on unseen views. Finally, we provide additional qualitative comparisons of indirect illumination during relighting in Figure 14, where our method produces more realistic, accurate indirect illumination than the baseline models.

Table 4: Quantitative comparison against baselines (IRGS, SVG-IR) and our ablation model on our new dataset. Our method significantly outperforms all baselines in indirect illumination reconstruction.

| Method | NVS | | | Indirect Illumination | | | Geometry | Albedo | | |
|---|---|---|---|---|---|---|---|---|---|---|
| | PSNR ↑ | SSIM ↑ | LPIPS ↓ | PSNR ↑ | SSIM ↑ | LPIPS ↓ | Normal MAE ↓ | PSNR ↑ | SSIM ↑ | LPIPS ↓ |
| IRGS | 35.0982 | 0.9660 | 0.0436 | 24.2219 | 0.8792 | 0.1092 | 4.1835 | 30.0931 | **0.9521** | 0.0752 |
| SVG-IR | 36.9634 | 0.9786 | 0.0266 | 30.9747 | 0.9134 | 0.0843 | 4.2624 | 29.7354 | 0.9309 | 0.0822 |
| Ours w/o $\mathcal{L}_{rad}$ | 36.3471 | 0.9764 | 0.0276 | 30.0954 | 0.9161 | 0.0752 | 3.8332 | 30.3425 | 0.9470 | 0.0760 |
| **Ours** | **37.8519** | **0.9822** | **0.0212** | **32.8832** | **0.9266** | **0.0726** | **3.6048** | **30.6224** | 0.9502 | **0.0744** |

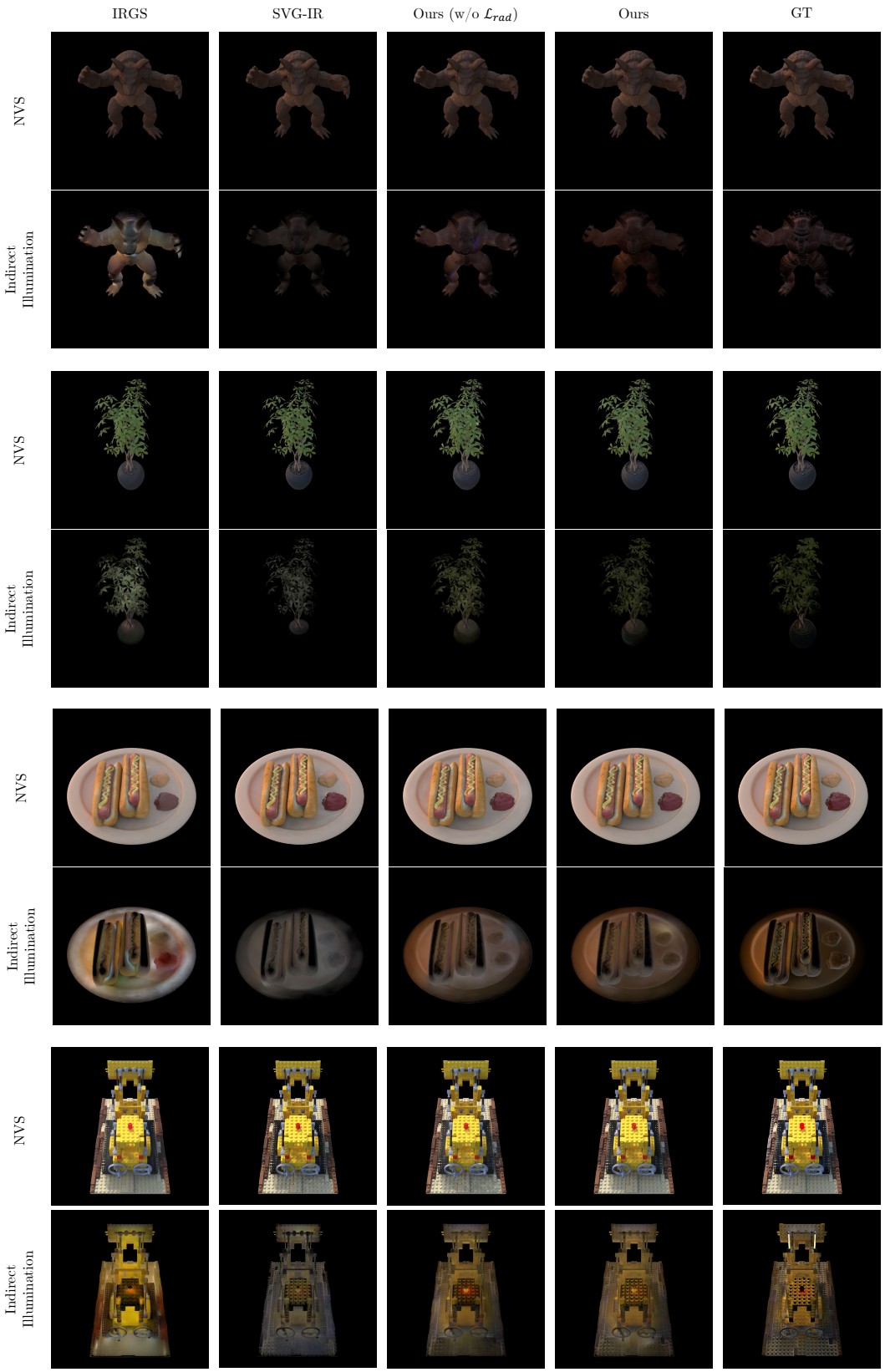

Figure 13: Qualitative comparison on novel-view synthesis and indirect illumination on our dataset. Best viewed in zoom.

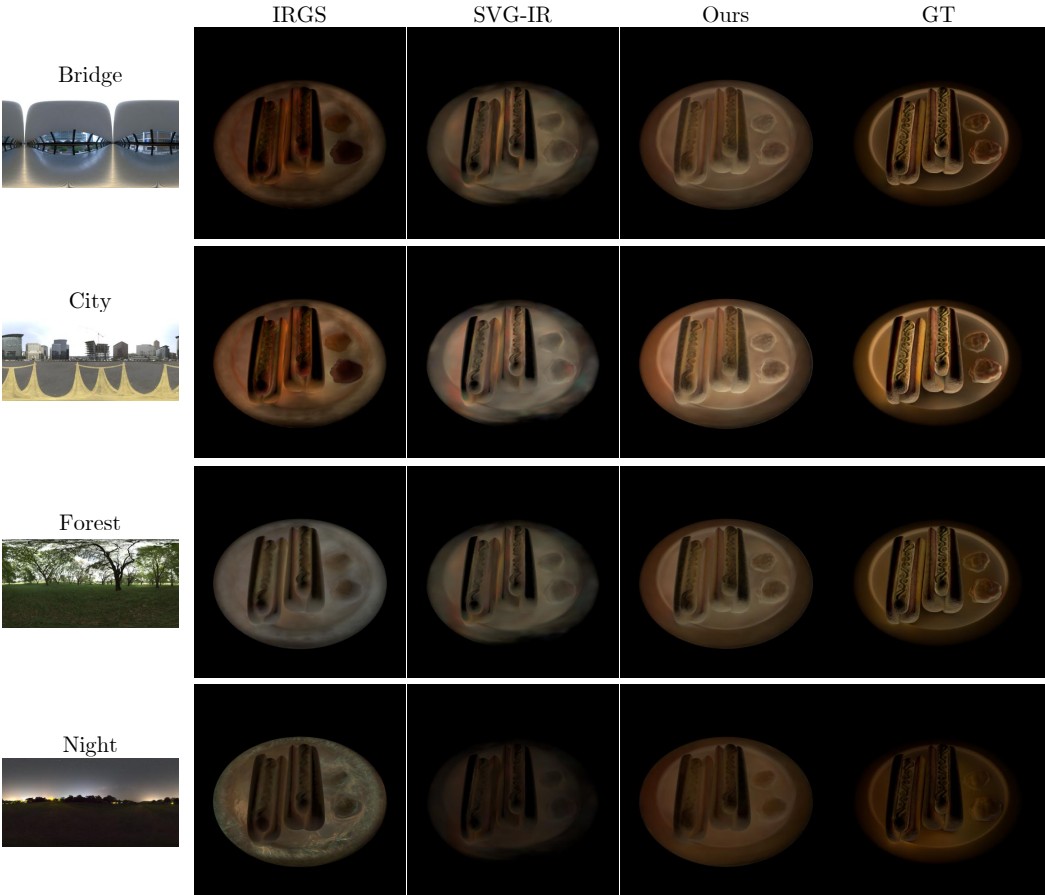

Figure 14: Qualitative comparison on indirect illumination for four different lighting conditions on the "hotdog" scene of our dataset. Best viewed in zoom.

# F    ABLATION STUDIES ON RADIOMETRIC CONSISTENCY

In this section, we discuss the contributions of the two main components of our framework, differentiable Gaussian ray tracing and supervision on unobserved (*i.e.* unseen) direction.

## F.1    RADIOMETRIC CONSISTENCY AND DIFFERENTIABLE GAUSSIAN RAY TRACING

The primary challenge in applying radiometric consistency to existing GS-based inverse rendering is that existing pipelines cannot dynamically query surfel radiances as indirect radiances during optimization. Existing pipelines employ point-based ray tracing or baked volumes, which precompute indirect radiance from NVS-pretrained Gaussian primitives in a non-differentiable manner. These precomputed values remain fixed during the optimization. When these values do not reflect updated Gaussian attributes, the supervision signal from radiometric consistency may become inconsistent. Our framework addresses this issue by employing differentiable Gaussian ray tracing to query indirect radiance for sampled surfels at every iteration.

We further conducted ablation studies to show how our framework enhances the contribution of radiometric consistency in Gaussian-based inverse rendering. Starting from the model initialized with our method, we trained three ablation models using different methods for querying indirect radiances and rendering, while utilizing our radiometric consistency loss during inverse rendering optimization.

- "Split-sum": Applies split-sum approximation to calculate physically-based radiance during the inverse rendering optimization, which does not involve indirect illumination.
- "RT precompute": Applies Monte Carlo estimate (Eq.(8)) to calculate and precomputes indirect radiances via ray tracing, freezing the indirect illumination estimate during the optimization.
- "RT w/o diff.": Dynamically updates indirect radiances via ray tracing during optimization, but without updating through ray-traced surfels by detaching the gradient on the ray-traced results.

The quantitative results of the three ablation models and our method are depicted in Table 5. "Split-sum" shows severe degradation in normal, albedo, and relighting accuracy due to the lack of illumination effects from surfels. "RT Precompute" shows enhanced normal, albedo, and relighting accuracy by utilizing precomputed indirect illumination, but yields the lowest novel-view synthesis performance among all methods. While "RT w/o Diff" improves overall performance through dynamic updates to these values, it still falls short of our method across all metrics. Ours method achieves the best performance across all metrics, demonstrating that the fully differentiable self-correcting feedback loop is essential for robust disentanglement.

Table 5: **Ablation study on radiometric consistency strategies.** We compare our method with baselines using different indirect illumination handling. Best results are highlighted in **bold**.

| Method | NVS | | | Geometry | Albedo | | | Relighting | | |
|---|---|---|---|---|---|---|---|---|---|---|
| | PSNR ↑ | SSIM ↑ | LPIPS ↓ | Normal MAE ↓ | PSNR ↑ | SSIM ↑ | LPIPS ↓ | PSNR ↑ | SSIM ↑ | LPIPS ↓ |
| Split-sum | 36.659 | 0.9800 | **0.0254** | 4.0066 | 27.471 | 0.9339 | 0.0828 | 28.816 | 0.9371 | 0.0575 |
| RT Precompute | 35.547 | 0.9728 | 0.0332 | 3.7138 | 30.523 | 0.9481 | 0.0750 | 31.965 | 0.9516 | 0.0496 |
| RT w/o Diff. | 37.252 | 0.9789 | 0.0275 | 3.6946 | 30.673 | 0.9507 | 0.0732 | 32.050 | 0.9524 | 0.0484 |
| **Ours** | **37.858** | **0.9801** | 0.0266 | **3.6889** | **31.048** | **0.9523** | **0.0719** | **32.092** | **0.9533** | **0.0478** |

## F.2    SUPERVISION ON UNOBSERVED DIRECTION

To further support our interpretation that the radiometric consistency supervises unseen directions, we conducted an additional ablation where we train both "Ours" and "Ours w/o $\mathcal{L}_{rad}$" on only 50% and 25% of the randomly subsampled training views of our new dataset on Appendix E. The numerical results on NVS and indirect illumination reconstruction performance are in Table 6. We have denoted the performance drop relative to the full training view (100%) on the right side of the metrics. NVS metrics degrade for both methods when fewer views are used. However, when using 25% of the training views, indirect illumination reconstruction with our model remains nearly

unchanged (-0.17dB), whereas the ablation shows a significant drop in indirect PSNR (-2.21dB). This indicates that when the camera viewpoint is limited, radiometric consistency still provides effective supervision that cannot be provided by NVS alone.

Table 6: **Ablation study on training view scarcity.** We report NVS and Indirect PSNR metrics across different subsets of training views. Values in parentheses denote the performance drop relative to the 100% setting.

| Train views | NVS PSNR | | Indirect PSNR | |
|---|---|---|---|---|
| | **Ours** | **Ours w/o $\mathcal{L}_{rad}$** | **Ours** | **Ours w/o $\mathcal{L}_{rad}$** |
| 100% | 37.85 | 36.35 | 32.88 | 30.10 |
| 50% | 37.54 (-0.31) | 35.81 (-0.54) | 32.79 (-0.09) | 29.15 (-0.95) |
| 25% | 36.79 (-1.06) | 34.65 (-1.70) | 32.71 (-0.17) | 27.89 (-2.21) |

# G  ABLATION STUDY ON HYPERPARAMETERS $N_g$ AND $N_s$

## G.1  ANALYSIS ON TENSOIR DATASET

We evaluated the performance impact of two key hyperparameters on the TensoIR dataset: the number of surfels sampled for radiometric consistency ($N_g$) and the number of incident ray samples ($N_s$) per surfel.

First, we varied $N_g$ from 1024 to 8192 while keeping $N_s$ fixed at 64. As shown in Table 7, increasing $N_g$ generally improves reconstruction quality across all tasks and metrics. This improvement is attributed to the radiometric consistency loss supervising a larger number of surfels per iteration. Notably, $N_g$ does not impact the rendering cost during inference, as it strictly controls the number of surfels supervised during the optimization step.

Table 7: Ablation study on the number of surfels ($N_g$) for radiometric consistency. We vary $N_g$ while fixing $N_s = 64$. Increasing $N_g$ improves quality without affecting rendering cost.

| $N_g$ | NVS | | | Geometry | Albedo | | | Relight | | | Render |
|---|---|---|---|---|---|---|---|---|---|---|---|
| | PSNR ↑ | SSIM ↑ | LPIPS ↓ | Normal MAE ↓ | PSNR ↑ | SSIM ↑ | LPIPS ↓ | PSNR ↑ | SSIM ↑ | LPIPS ↓ | (ms) |
| 1024 | 37.8206 | 0.9799 | 0.0268 | 3.6900 | 30.9137 | 0.9518 | 0.0730 | 32.0569 | 0.9529 | 0.0481 | 38.5 |
| 2048 (Ours) | 37.8580 | 0.9801 | 0.0266 | 3.6889 | 31.0479 | 0.9521 | 0.0721 | 32.0920 | 0.9533 | 0.0478 | 38.6 |
| 4096 | 37.8707 | 0.9802 | 0.0264 | 3.6852 | 31.0495 | 0.9522 | 0.0721 | 32.1112 | 0.9532 | 0.0478 | 38.7 |
| 8192 | 37.8799 | 0.9802 | 0.0263 | 3.6834 | 31.0507 | 0.9523 | 0.0720 | 32.1294 | 0.9533 | 0.0477 | 38.5 |

Next, we analyzed the effect of $N_s$ ranging from 16 to 128 with $N_g$ fixed at 2048. Table 8 demonstrates that reconstruction quality improves as $N_s$ increases up to 64. However, increasing $N_s$ further to 128 yields diminishing returns, and in some tasks (e.g., NVS and Relighting), performance slightly drops. This suggests that the additional ray samples beyond this point do not significantly resolve the variance in Monte Carlo integration for the given capacity. regarding efficiency, the rendering cost scales with $N_s$ due to the additional computation required for the Monte Carlo estimate of $L_G^{PBR}$. However, the rendering cost remains manageable even at $N_s = 128$, achieving 58.3 ms per frame ($\sim$17.2 fps).

Table 8: Ablation study on the number of secondary ray samples ($N_s$). We vary $N_s$ while fixing $N_g = 2048$. Performance saturates around $N_s = 64$.

| $N_s$ | NVS | | | Geometry | Albedo | | | Relight | | | Render |
|---|---|---|---|---|---|---|---|---|---|---|---|
| | PSNR ↑ | SSIM ↑ | LPIPS ↓ | Normal MAE ↓ | PSNR ↑ | SSIM ↑ | LPIPS ↓ | PSNR ↑ | SSIM ↑ | LPIPS ↓ | (ms) |
| 16 | 37.4199 | 0.9792 | 0.0275 | 3.7174 | 30.8656 | 0.9505 | 0.0735 | 32.1684 | 0.9518 | 0.0494 | 16.2 |
| 32 | 37.8125 | 0.9799 | 0.0268 | 3.6979 | 30.9014 | 0.9510 | 0.0730 | 32.1537 | 0.9528 | 0.0483 | 22.6 |
| 64 (Ours) | 37.8707 | 0.9802 | 0.0264 | 3.6852 | 31.0479 | 0.9521 | 0.0721 | 32.1112 | 0.9532 | 0.0478 | 38.6 |
| 128 | 37.6548 | 0.9798 | 0.0268 | 3.6821 | 30.9563 | 0.9512 | 0.0728 | 31.9768 | 0.9526 | 0.0480 | 58.3 |

## G.2  RENDERING EFFICIENCY ON LARGE-SCALE SCENES (MIPNERF360)

We further provide analysis on MipNeRF360 (Barron et al., 2022) dataset to evaluate the rendering cost in complex, scene-level environments. Qualitative results corresponding to this analysis are provided in Figure 15.

We further investigated the computational cost by varying $N_g$ from 1024 to 16384 while keeping $N_s$ fixed at 64 (Table 9). Consistent with the analysis on the TensoIR dataset, varying $N_g$ has a negligible impact on the rendering cost.

Finally, we analyzed the effect of the number of incident ray samples ($N_s$) on rendering time, varying $N_s$ from 16 to 64 with $N_g$ fixed at 2048 (Table 10). As shown, the rendering cost scales with $N_s$. This indicates that while larger scenes increase the baseline computational load, the rendering speed can be effectively controlled by adjusting $N_s$. Although reducing $N_s$ improves speed, it may trade off rendering quality. However, given that increasing $N_g$ improves quality without computational overhead, our method can scale to larger scenes while maintaining efficiency by strategically balancing $N_g$ and $N_s$.

Table 9: Rendering cost analysis on MipNeRF360 with varying $N_g$. We fix $N_s = 64$. Consistent with TensoIR, $N_g$ does not significantly affect rendering speed.

| $N_g$ | Average (ms) | Bonsai (ms) | Counter (ms) | Kitchen (ms) | Room (ms) |
|---|---|---|---|---|---|
| 1024 | 81.23 | 77.37 | 72.38 | 86.81 | 88.43 |
| 2048 (Ours) | 81.40 | 77.78 | 72.95 | 84.29 | 90.56 |
| 4096 | 81.69 | 78.87 | 72.19 | 85.28 | 90.41 |
| 8192 | 81.53 | 77.75 | 73.34 | 85.22 | 89.82 |
| 16384 | 81.86 | 77.75 | 73.24 | 86.08 | 90.38 |

Table 10: Rendering cost analysis on MipNeRF360 with varying $N_s$. We fix $N_g = 2048$. Reducing $N_s$ significantly decreases rendering time, allowing for trade-offs between speed and quality.

| $N_s$ | Average (ms) | Bonsai (ms) | Counter (ms) | Kitchen (ms) | Room (ms) |
|---|---|---|---|---|---|
| 16 | 24.57 | 23.49 | 23.86 | 26.07 | 24.85 |
| 32 | 43.43 | 42.00 | 39.32 | 47.31 | 45.10 |
| 64 (Ours) | 81.40 | 77.78 | 72.95 | 84.29 | 90.56 |

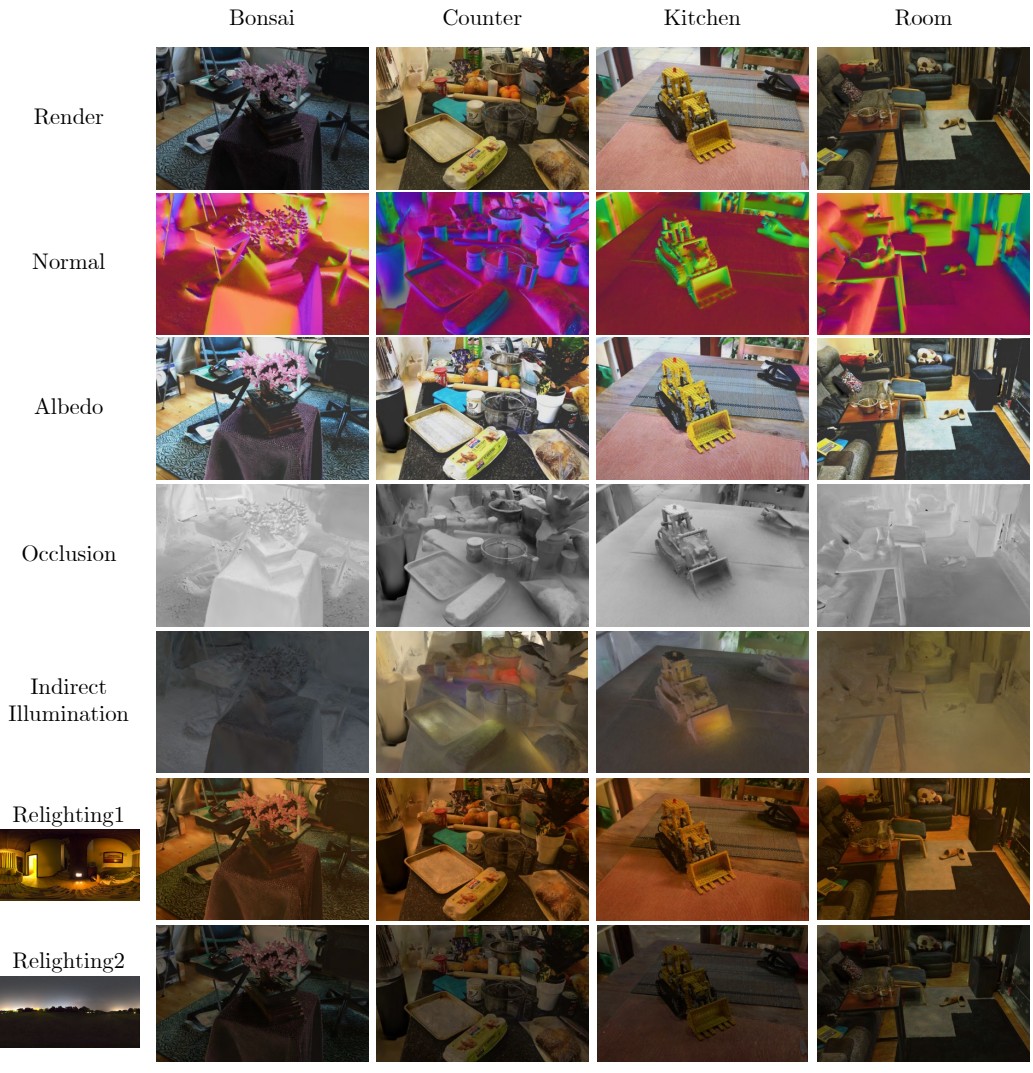

Figure 15: Qualitative results on the MipNeRF360 dataset. Best viewed in zoom.

# H  ADDITIONAL ANALYSIS ON THE INITIALIZATION STRATEGY

We compared our initialization strategy against NVS-based initialization methods. We define "NVS Init." as a model utilizing NVS pre-training for initialization followed by our full inverse rendering optimization. Additionally, we evaluated "NVS Init. w/o $\mathcal{L}_{rad}$", where the radiometric consistency loss is removed throughout both the initialization and the subsequent optimization stages.

Table 11 summarizes the quantitative results. The "NVS Init. w/o $\mathcal{L}_{rad}$" baseline exhibits the lowest performance across all tasks, highlighting the necessity of radiometric supervision. While introducing radiometric consistency after NVS pre-training ("NVS Init.") improves albedo and relighting quality, it still falls short of our method. Notably, even when our method is trained without radiometric consistency during the main optimization phase ("Ours w/o $\mathcal{L}_{rad}$"), it outperforms the full NVS-initialized model in albedo reconstruction. This suggests that enforcing physical constraints from the start is crucial for robust disentanglement.

Table 11: Ablation study on initialization strategies. We compare our initialization (Ours) against NVS-based initialization baselines. "NVS init." denotes NVS pre-training followed by our standard optimization. "during IR" refers to the inverse rendering optimization stage.

| Method | NVS PSNR ↑ | Albedo PSNR ↑ | Relight PSNR ↑ |
|---|---|---|---|
| NVS init. + $\lambda_{rad} = 0$ during IR | 35.70 | 30.31 | 31.51 |
| NVS init. | 37.43 | 30.34 | 31.66 |
| Ours w/o $\mathcal{L}_{rad}$ ($\lambda_{rad} = 0$ during IR) | 35.82 | 30.82 | 31.69 |
| **Ours** | **37.86** | **31.05** | **32.09** |

We further analyze the convergence behavior and qualitative results in Figure 16. As shown in the convergence plot (left of Figure 16), our initialization leads to substantially faster and more stable convergence of the radiometric consistency loss $\mathcal{L}_{rad}$, whereas standard NVS initialization results in higher and noisier residuals. This demonstrates that conventional NVS initialization places the surfels into a suboptimal orientation that is misaligned with the physical decomposition required for inverse rendering with radiometric consistency. Qualitatively (right of Figure 16), NVS pre-training causes the model to misinterpret geometric cues, leading to shadows baked into the geometry (red arrows) and inter-reflections merged into the albedo (blue arrows). In contrast, our initialization preserves physical priors from the beginning, enabling enhanced geometry reconstruction and a cleaner separation of inter-reflection effects.

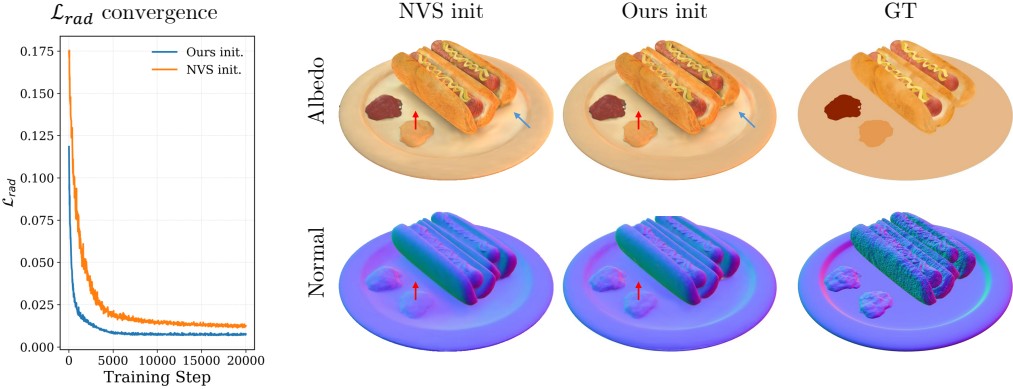

Figure 16: Ablation study of our initialization method on the "hotdog" scene of TensoIR dataset.

# I  ENVIRONMENT MAP RECONSTRUCTION ON BENCHMARK DATASETS

We visualized the optimized environment maps from the TensoIR (Jin et al., 2023) and Synthetic4Relight (Zhang et al., 2022) datasets and compared them with baseline methods and the Ground Truth (GT) in Figures 17 and 18. Our method consistently recovers environment maps that are visually closest to the ground truth environment map, reflecting the benefit of our radiometric consistency, leading to robust disentanglement of illumination for inverse rendering. In contrast, GI-GS and GS-IR show unnatural environment maps compared to other baseline methods. Baselines relying on point-based ray tracing (R3DG and SVG-IR) often fail to disentangle base color from the environment map, such as for the scenes "hotdog" and "lego" from TensoIR, and "air balloons" from the Synthetic4Relight dataset. While IRGS uses differentiable Gaussian ray tracing, it tends to overestimate the intensity in regions that should be dark, such as the ground of the GT environment map.

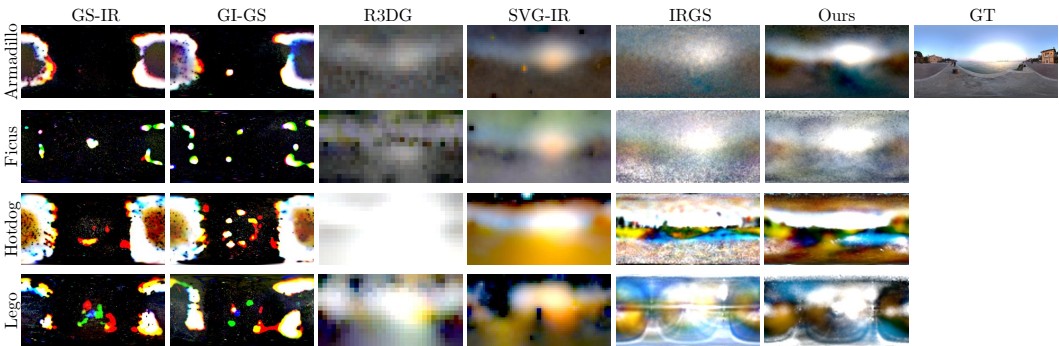

Figure 17: Qualitative comparison on reconstructed environment map on TensoIR dataset. Best viewed in zoom.

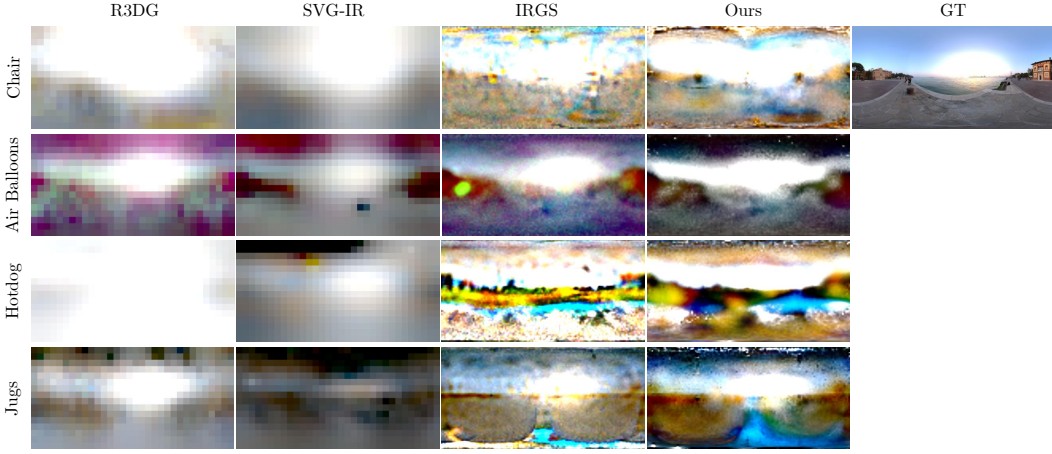

Figure 18: Qualitative comparison on reconstructed environment map on Synthetic4Relight dataset. Best viewed in zoom.

## J ADDITIONAL ANALYSIS ON FINETUNING-BASED RELIGHTING STRATEGY

We visualize the convergence and visual progression in Figure 19. As finetuning progresses, the image rendered by surfel radiances quickly converges towards the PBR reference (orange line, left plot). However, the quality of the PBR reference itself slightly degrades over time (blue line, left plot), due to the errors accumulated by the finetune surfel radiances that contribute as indirect illumination for physically-based rendering. Nevertheless, our method quickly adapts surfel radiances to new lighting conditions, allowing us to directly use learned surfel radiances to rasterize relighted frames. We additionally provide the corresponding visual comparisons in Figure 20 of the revised paper, which demonstrates that our finetuning-based relighting method shows similar relighting quality compared to ray-traced results.

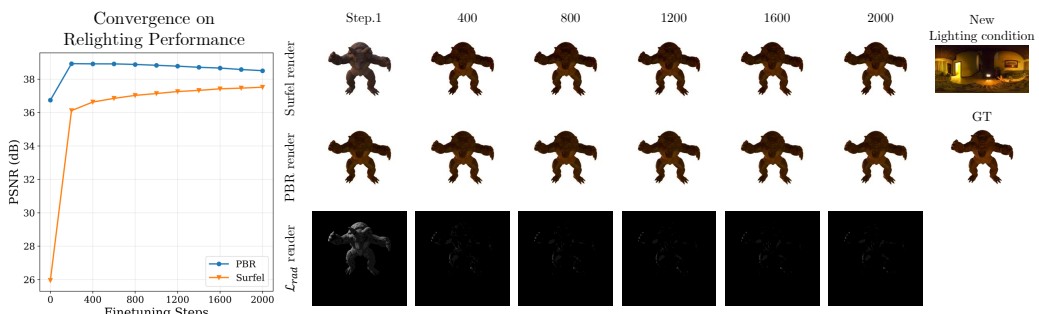

Figure 19: Illustrative figure on how our finetuning-based relighting adapts surfel radiances for new lighting conditions.

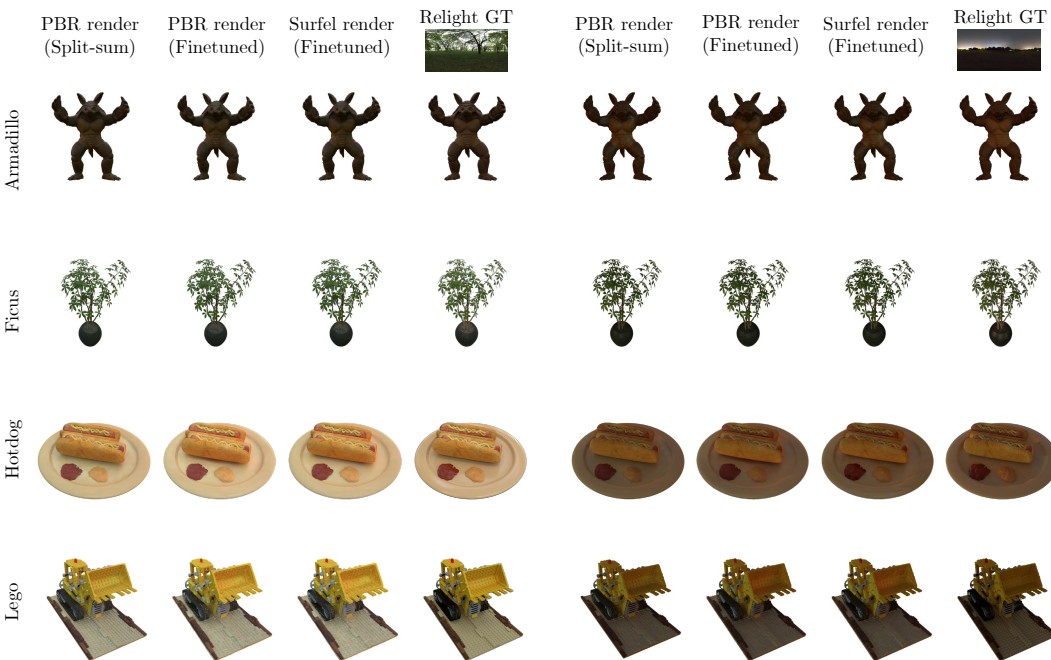

Figure 20: Qualitative ablation study on three relighting method: 1) Gaussian ray tracing that queries indirect radiance via split-sum approximation "PBR render (Split-sum)", 2) Gaussian ray tracing that queries indirect radiance from finetuned surfels "PBR render (Finetuned)", and 3) direct rasterization with finetuned surfel radiances "Surfel render (Finetuned)".

While our proposed finetuning-based relighting introduces a pre-computation step compared to existing relighting pipelines, it offers advantages in terms of rendering cost and memory efficiency.

We analyze the trade-off between additional finetuning cost and real-time capabilities by comparing average rendering speed and VRAM consumption on the TensoIR dataset. We compare our finetuned model against a baseline relighting method that utilizes Gaussian ray tracing for shading with varying sample counts $N_s \in \{64, 128, 256\}$.

As reported in Table 12, although finetuning incurs an upfront computational cost, it eliminates the necessity of storing incident radiance for every surfel during inference. This results in significantly reduced memory usage and faster inference speeds ($\sim$5.90 ms). In contrast, baseline relighting methods demonstrate increasing rendering time and memory consumption proportional to the sample count $N$, as they require storing dense incident ray information for shading each surfel. These results suggest that our method is particularly suitable for memory-constrained environments, such as consumer-grade GPUs or edge devices, where storing dense lighting data becomes prohibitive.

Table 12: Comparison of rendering speed and memory usage. We compare our finetuned model against the baseline relighting method with varying ray tracing sample counts ($N$). Our method achieves significantly lower rendering time and memory footprint.

| Method | Sample Count ($N$) | Rendering (ms) ↓ | VRAM (MB) ↓ |
|---|---|---|---|
| **Ours (Finetuned)** | - | **5.90** | **308.2** |
| Baseline (Ray Tracing) | 64 | 38.64 | 1512.6 |
| | 128 | 58.31 | 2523.3 |
| | 256 | 86.99 | 4589.5 |

