# OpenReview forum: "Radiometrically Consistent Gaussian Surfels for Inverse Rendering"
_ICLR.cc/2026/Conference — ICLR 2026 Oral_

### Official Review · Reviewer_ShdJ · 2025-10-28

**Soundness:** 3
**Presentation:** 3
**Contribution:** 3
**Rating:** 6
**Confidence:** 4

**Summary:**

The paper proposes a physically grounded inverse rendering framework called RadioGS. It introduces radiometric consistency, a physical constraint that enforces the agreement between each Gaussian surfel’s learned radiance and its physically-based rendered counterpart. This self-correcting mechanism provides supervision for unobserved views, enabling accurate modeling of indirect illumination and inter-reflections. Built on 2D Gaussian ray tracing, RadioGS integrates this constraint efficiently to improve decomposition of geometry, material, and lighting. Moreover, it includes a finetuning-based relighting strategy that adapts surfel radiances to new lighting conditions within minutes while maintaining real-time rendering speed (<10 ms). Experiments on benchmark datasets demonstrate that RadioGS achieves superior inverse rendering and relighting quality over previous Gaussian-based and NeRF-based methods

**Strengths:**

1. The introduction of radiometric consistency provides principled physical supervision for unobserved views.

2. The proposed finetuning-based relighting method allows adaptation to new lighting conditions within minutes, achieving high-quality results with less than 10 ms rendering time per frame.

3. Extensive experiments on multiple benchmarks (TensoIR, Synthetic4Relight, Stanford-ORB) demonstrate superior performance in both inverse rendering and relighting tasks compared to prior Gaussian and NeRF-based methods.

**Weaknesses:**

1.The contribution may be limited in novelty, the proposed radiometric consistency mainly enforces agreement between the surfel’s learned radiance and its physically-based rendered results, which is a relatively straightforward physical constraint.

2.The relighting process requires additional finetuning under the proposed constraint, which introduces extra computation before rendering and reduces its advantage in fully real-time applications.

3.Although the paper claims that the finetuning stage improves both realism and efficiency of relighting, the experiments lack an ablation study that explicitly compares relighting quality with and without the finetuning stage and also the physically-based rendering component.

**Questions:**

See weakness.

---

> ### Author Response · Authors · 2025-11-21
> **Response to Reviewer ShdJ (1/3)**
>
> ### W1. Novelty on Radiometric Consistency
>
> We thank the reviewer for the feedback on our novelty regarding the radiometric consistency. The principle of enforcing consistency with the rendering equation is straightforward. However, we would like to emphasize that effectively utilizing radiometric consistency for Gaussian Splatting (GS)-based inverse rendering pipeline poses a challenge, and our framework addresses the challenge.
>
> Applying radiometric consistency to existing GS-based inverse rendering for enhanced indirect illumination is difficult because existing GS-based inverse rendering pipelines cannot query dynamically updated surfel radiances as indirect radiances during optimization. Existing GS-based inverse rendering methods employ point-based ray tracing, which precomputes indirect radiance from NVS-pretrained Gaussian primitives in a non-differentiable manner. These precomputed values remain fixed during the optimization, and thus fail to reflect updated Gaussian attributes. This may lead the supervision signal from radiometric consistency to become inconsistent.
>
> Our framework addresses this issue by employing differentiable Gaussian ray tracing to query indirect radiance for sampled surfels at every iteration. We further conducted ablation studies to show how our framework enhances the contribution of radiometric consistency for Gaussian-based inverse rendering. Starting from the model initialized with our method, we trained three ablation models with different methods in querying indirect radiances and rendering while utilizing our radiometric consistency loss during inverse rendering optimization.
>
> - "Split-sum": Applies split-sum approximation to calculate physically-based radiance $L_G^{PBR}$ during the inverse rendering optimization, which does not involve indirect illumination.
> - "RT precompute": Applies Monte Carlo estimate (Eq.(8)) to calculate $L_G^{PBR}$ and precomputes indirect radiances via ray tracing, blocking further update on indirect illumination during the optimization.
> - "RT w/o diff.": Dynamically updates indirect radiances via ray tracing during optimization, but without update through ray-traced surfels by detaching the gradient on the ray-traced results.
>
> The results of the three ablation models and our method are depicted in the table below. "Split-sum" shows severe degradation in normal, albedo, and relighting accuracy due to the lack of illumination effects induced by the surfels. "RT Precompute" shows enhanced normal, albedo, and relighting accuracy by utilizing precomputed indirect illumination, but yields the lowest novel-view synthesis performance among all methods. While "RT w/o Diff" improves overall performance via dynamic updates on these values, it still falls short of our method in all metrics. "Ours" achieves the best performance across all metrics, demonstrating that the fully differentiable self-correcting feedback loop is essential for robust disentanglement.
>
> | Method         | NVS PSNR ↑ | NVS SSIM ↑ | NVS LPIPS ↓ | Normal MAE ↓ | Albedo PSNR ↑ | Albedo SSIM ↑ | Albedo LPIPS ↓ | Relight PSNR ↑ | Relight SSIM ↑ | Relight LPIPS ↓ |
> |----------------|------------|------------|--------------|---------------|----------------|----------------|-----------------|-----------------|-----------------|------------------|
> | Split-sum      | 36.659     | 0.9800     | **0.0254**   | 4.0066        | 27.471         | 0.9339         | 0.0828          | 28.816          | 0.9371          | 0.0575           |
> | RT precompute  | 35.547     | 0.9728     | 0.0332       | 3.7138        | 30.523         | 0.9481         | 0.0750          | 31.965          | 0.9516          | 0.0496           |
> | RT w/o diff    | 37.252     | 0.9789     | 0.0275       | 3.6946        | 30.673         | 0.9507         | 0.0732          | 32.050          | 0.9524          | 0.0484           |
> | Ours        | **37.858** | **0.9801** | 0.0266       | **3.6889**    | **31.048**     | **0.9523**     | **0.0719**      | **32.092**      | **0.9533**      |  **0.0478**       |
>
> In summary, we would like to emphasize that our contribution is to seamlessly integrate radiometric consistency to Gaussian-based inverse rendering, allowing Gaussian surfel radiances on unseen directions to be supervised based on the radiometric consistency.

---

> ### Author Response · Authors · 2025-11-21
> **Response to Reviewer ShdJ (2/3)**
>
> ### W2. Precomputation for Finetuning-based Relighting
>
> We appreciate the reviewer's comment regarding the trade-off between the additional finetuning cost and real-time capabilities. We agree that the proposed finetuning-based relighting introduces a pre-computation step, which presents a limitation regarding real-time applications. To address your concern, we conducted an additional analysis comparing the average rendering speed and VRAM consumption between our finetuned model and the baseline relighting method using Gaussian ray tracing for shading, with varying sample counts $N_s$ from $64$ to $256$. All experiments were conducted on the TensoIR dataset.
>
> As reported in the table below, while finetuning incurs an upfront computational cost, it eliminates the need to store incident radiance for every surfel. This leads to reduced memory usage and faster inference speeds (~5.9 ms) during the actual rendering phase. Baseline relighting methods show increasing rendering time and memory consumption proportional to $N_s$.
>
> | **Method** | **Rendering (ms)** | **VRAM consumption (MB)** |
> | --- | --- | --- |
> | finetuned | 5.90 | 308.2 |
> | $N_s=64$ | 38.64 | 1512.6 |
> | $N_s=128$ | 58.31 | 2523.3 |
> | $N_s=256$ | 86.99 | 4589.5 |
>
> We believe this characteristic may extend the applicability of our method to memory-constrained environments (e.g., consumer-grade GPUs or edge devices) where storing dense incident ray information for rendering becomes prohibitive.
>
> __________________________________________________________________________________________________
> ### W3. Additional Analysis and Comparison on Finetuning-based Relighting
>
> We would like to clarify a potential misunderstanding regarding our claims about the finetuning-based relighting strategy. We do not claim that the finetuning-based method achieves superior visual quality compared to the full ray-tracing-based PBR method. Instead, the primary goal of the finetuning stage is to enable real-time rendering speeds under new lighting conditions through efficient precomputation. As discussed in Sec 5.2 of our submitted paper, the finetuning process accumulates minor errors from the estimated geometry and material properties into the surfel radiances. Consequently, while the surfel radiances successfully adapt to the new lighting, there is a slight trade-off in rendering quality compared to the physically-based rendering method.
>
> We provide a convergence analysis and visual progression of our finetuning-based relighting method in Figure 19 of the revised paper. As the finetuning progresses, the relighted image rendered by the surfel radiances quickly converges towards the physically-based rendered image (orange line of the left plot). However, the quality of the physically-based rendered image slightly degrades as the finetuning progresses (blue line of the left plot), due to the errors accumulated by the finetune surfel radiances that contribute as indirect illumination for physically-based rendering. Nevertheless, our method quickly adapts surfel radiances to new lighting conditions, allowing us to directly use learned surfel radiances to rasterize relighted frames.
>
> To quantitatively demonstrate the trade-off between rendering time and relighting quality, we have conducted an ablation study on the TensoIR dataset using the following three configurations:
>
> - "PBR render (Split-sum)": Gaussian ray tracing that estimates indirect radiance using the PBR split-sum approximation in ray-traced surfaces.
> - "PBR render (Finetuned)": Gaussian ray tracing that uses indirect radiance queried by finetuned surfels.
> - "Surfel render (Finetuned)": Direct rasterization using finetuned surfel radiances.

---

> > ### Author Response · Authors · 2025-12-03
> >
> > Dear Reviewer SdhJ,
> >
> > We apologize for the oversight in reporting VRAM consumption, which incorrectly included memory usage during dataset loading.
> >
> > We have updated the comment and the paper with the corrected VRAM consumption metric, reflecting only the memory utilized for rendering with our RadioGS method.
> >
> > Please refer to the updated metrics and Appendix J, Table 12 in the revised paper.
> >
> > Sincerely,
> >
> > Authors.

---

> ### Author Response · Authors · 2025-11-21
> **Response to Reviewer ShdJ (3/3)**
>
> As shown in the table below, our "Surfel render (Finetuned)" method achieves a huge reduction in rendering time compared to the PBR methods, with the cost of a decrease in quality. Yet, we would like to emphasize that our Surfel render (Finetuned) still outperforms the baseline methods in terms of relighting quality. Comparison between "PBR render (Split-sum)" and "PBR render (Finetuned)" also demonstrates the minor quality trade-off, where the performance degradation is less severe since the finetuned surfel radiances contribute as indirect radiance for physically-based rendering. We provide the corresponding visual comparisons in Figure 20 of the revised paper, which demonstrates that our finetuning-based relighting method shows similar relighting quality compared to ray-traced results.
>
> | Method | PSNR $\uparrow$ | SSIM $\uparrow$ | LPIPS $\downarrow$ | Rendering (ms) |
> | :---: | :---: | :---: | :---: | :---: |
> | IRGS | 29.91 | 0.935 | 0.076 | 1090 |
> | SVG-IR | 31.10 | 0.946 | 0.056 | 82.49 |
> | Surfel render (Finetuned) | 31.41 | 0.948 | 0.052 | 5.90 |
> | PBR render (Finetuned) | 31.59 | 0.952 | 0.049 | 38.29 |
> | PBR render (Split-sum) | 32.09 | 0.953 | 0.048 | 38.64 |

---

> ### Author Response · Authors · 2025-11-27
>
> Dear reviewer ShdJ,
>
> We sincerely appreciate for taking the time to review our paper and to provide insightful feedback.
> This is a gentle reminder that the discussion is scheduled to conclude on December 2nd.
> We kindly ask whether the concerns have been sufficiently addressed.
> We would be grateful to discuss any remaining questions and concerns the reviewer may have before the deadline.

---

### Official Review · Reviewer_47iG · 2025-10-28

**Soundness:** 3
**Presentation:** 3
**Contribution:** 2
**Rating:** 6
**Confidence:** 5

**Summary:**

This paper presents a novel inverse rendering framework, RadioGS, which better modeling the complex material-lighting interaction by utilizing a novel radiometric consistency loss. The proposed radiometric consistency loss are built on the 2D Gaussian Ray Tracing and enables more accurate indirect illumination querying. Experiment results are provided to demonstrate the effectiveness of the framework, highlighting the effectiveness of the novel loss.

**Strengths:**

1. The paper propose a novel inverse rendering framework where 2DGRT are used to calculate the rendering equation and radiometric consistency loss, which achieve accurate inter-reflection modeling and material estimation

2. The proposed initialization and finetune-based relighting schemes further improve the inverse rendering and relighting performance.

3. Extensive experiments show that the method achieve best performance over inverse rendering baselines.

**Weaknesses:**

1. The proposed loss ensures the consistency between the surfel’s outgoing radiance and the PBR result. But it cannot promise that the outgoing radiance $L_G$ is correct in unseen direction. So I'm curious why it can improve the inverse rendering performance.
2. As we are talking about inverse rendering, so the material estimation should be more important than NVS. But according to the albedo PSNR shown in Fig.6, it seems that the performance gain comes more from NVS init. This will weaken the core contribution of the paper.

**Questions:**

1. What does the optimized envmap looklike, as the indirect illumination is more accurate, the estimated direct illumination should also be better.
2. Why can a simple finetune make the surfel’s outgoing radiance adopt to new lighting conditions? It sounds magic~

---

> ### Author Response · Authors · 2025-11-21
> **Response to Reviewer 47iG (1/3)**
>
> ### W1. Regarding the Supervision Towards Unseen Direction
>
> We thank the reviewer for giving us the opportunity to clarify the core mechanism of our contribution. Our radiometric consistency loss $\mathcal{L}_{rad}$ provides supervision to learned surfel radiance $L_G$ on unseen direction by forcing to match the physically-based rendered radiance $L_G^{PBR}$. This leads $L_G$ to be consistent and physically induced by the scene's current estimated properties (geometry, material, lighting) for unseen directions not supervised by NVS training. By replacing an arbitrary radiance obtained from novel-view synthesis with a physically induced radiance from our method, the surfels become a relatively robust source of indirect illumination for inverse rendering.
>
> To empirically verify whether our method leads to accurate indirect illumination modeling, we conducted an additional experiment on a new dataset with ground truth (GT) indirect illumination. Since existing inverse rendering benchmarks do not provide GT indirect illumination, we utilized the original Blender files from the TensoIR dataset to generate ground truth indirect illumination. We used Blender Cycles path tracing engine to render the indirect illumination pass along with the original render pass. To ensure noise-free references, we set the sampling rate to 256 samples per pixel (spp) and applied the OIDN (Open Image Denoise)[1] denoiser. We trained our method and the ablation model (Ours w/o $\mathcal{L}{rad}$) with hyperparameters from the submitted paper, and also trained baseline models IRGS and SVG-IR on our new dataset for comparison. We evaluated rendered image, indirect illumination, normal, and albedo across the scenes.
>
> The table below presents the quantitative comparison against baselines (IRGS, SVG-IR) and our ablation model. Our method significantly outperforms all baselines in indirect illumination reconstruction, confirming that our method accurately models the physical transport of indirect light. When radiometric consistency is removed (Ours w/o $\mathcal{L}_{rad}$), the indirect PSNR drops significantly, showing that the performance gain comes from radiometric consistency. By correctly modeling the indirect illumination, our method shows superior performance in most metrics. The visual comparisons on indirect illumination are also presented in Figures 13 and 14 in the revised paper, where our method models accurate inter-reflections compared to the baseline and the ablation model across all four scenes, and also under new lighting conditions.
>
> | Method        | NVS PSNR ↑ | NVS SSIM ↑ | NVS LPIPS ↓ | Indirect PSNR ↑ | Indirect SSIM ↑ | Indirect LPIPS ↓ | Normal MAE ↓ | Albedo PSNR ↑ | Albedo SSIM ↑ | Albedo LPIPS ↓ |
> |---------------|------------|------------|--------------|------------------|------------------|-------------------|--------------|----------------|----------------|-----------------|
> | IRGS          | 35.0982    | 0.9660     | 0.0436       | 24.2219          | 0.8792           | 0.1092            | 4.1835       | 30.0931        | **0.9521**         | 0.0752          |
> | SVGIR         | 36.9634    | 0.9786     | 0.0266       | 30.9747          | 0.9134           | 0.0843            | 4.2624       | 29.7354        | 0.9309         | 0.0822          |
> | Ours w/o $\mathcal{L}_{rad}$  | 36.3471    | 0.9764     | 0.0276       | 30.0954          | 0.9161           | 0.0752            | 3.8332       | 30.3425        | 0.9470         | 0.0760          |
> | Ours       | **37.8519** | **0.9822** | **0.0212**    | **32.8832**       | **0.9266**        | **0.0726**         | **3.6048**    | **30.6224**     | 0.9502       | **0.0744**        |

---

> ### Author Response · Authors · 2025-11-21
> **Response to Reviewer 47iG (2/3)**
>
> ### W2. Ablation on Initialization Methods
>
> We appreciate the reviewer for the opportunity to clarify the ablation settings and the specific contributions of each component. We apologize for the ambiguity in our submitted paper.
> "NVS init." of the table of Figure 6 of our submitted paper refers to an ablation model utilizing NVS pre-training for initialization, but still applying our inverse rendering optimization. The albedo reconstruction performance is still moderate due to our radiometric consistency loss during the inverse rendering optimization, but is yet inferior compared to utilizing our initialization method. Our additional Figure 16 on the revised paper further demonstrates visual comparison between our method and the "NVS init". Our method shows enhanced reconstruction on geometry (red arrow) and albedo reconstruction (blue arrow), while "NVS init" bakes shadows into geometry (red arrow) and inter-reflections to albedo (blue arrow).
> We have conducted an additional ablation study by removing our radiometric consistency loss throughout the whole optimization (NVS init. + $\lambda_{\text{rad}}=0$ during IR). As demonstrated in the first row of the table below, this led to further deteriorated performance on all three tasks. Our additional ablation study reflects the contribution of our radiometric consistency loss on both initialization and inverse rendering stages.
>
> | Method | NVS PSNR $\uparrow$ | Albedo PSNR $\uparrow$ | Relight PSNR $\uparrow$ |
> | :---: | :---: | :---: | :---: |
> | NVS init. + $\lambda_{\text{rad}}=0$ during IR | 35.70 | 30.31 | 31.51 |
> | NVS init. | 37.43 | 30.34 | 31.66 |
> | $\lambda_{\text{rad}}=0$ during IR | 35.82 | 30.82 | 31.69 |
> | Ours | **37.86** | **31.05** | **32.09** |
>
> __________________________________________________________________________________________________
> ### Q1. Visualization on Environment Maps
>
> We have visualized the optimized environment maps from the TensoIR and Synthetic4Relight datasets and compared them with baseline methods and the Ground Truth (GT) in Figures 17 and 18 of the revised paper. We observed that our method consistently recovers environment maps that are visually closest to the ground truth environment map, reflecting the benefit of our radiometric consistency, leading to robust disentanglement of illumination for inverse rendering. In contrast, GI-GS and GS-IR show unnatural environment maps compared to other baseline methods. Baselines relying on point-based ray tracing (R3DG and SVG-IR) often fail to disentangle base color from the environment map, such as for the scenes “hotdog” and “lego” from TensoIR, and “air balloons” from the Synthetic4Relight dataset. While IRGS uses differentiable Gaussian ray tracing, it tends to overestimate the intensity in regions that should be dark, such as the ground of the GT environment map.

---

> ### Author Response · Authors · 2025-11-21
> **Response to Reviewer 47iG (3/3)**
>
> ### Q2. Mechanism of Finetuning-based Relighting
>
> We are grateful for the chance to elaborate on our finetuning-based relighting method. Our method aligns with the principle of self-training neural radiance caches used in dynamic scene rendering [2,3,4]. Similarly, our method updates the surfel radiance to match the new lighting condition. To clarify this, we provide a convergence analysis and visual progression of our finetuning-based relighting method in Figure 19 of the revised paper.
>
> The efficacy of our method comes from the self-correcting feedback between the physically-based rendered (PBR) radiance $L_G^{PBR}$ and learned surfel radiance $L_G$ to reduce the radiometric consistency loss $\mathcal{L}{rad}$ on a new lighting condition.
>
> Firstly, as shown in the blue line (PBR) of the plot, the PBR rendered image achieves a relatively high PSNR immediately upon changing the lighting condition. This is because the PBR radiance $L_G^{PBR}$ explicitly queries direct illumination from the new lighting condition and estimates the radiance based on the current geometry and material. On the other hand, the orange line (Surfel) of the plot, the Surfel-rendered image shows low PSNR since the surfel radiance $L_G$ is optimized for the previous lighting condition. Thus, PBR radiance  $L_G^{PBR}$ serves as a physically grounded target for optimizing the surfel radiance $L_G$ towards $L_G^{PBR}$.
>
> Secondly, as the surfel radiance $L_G$ gets refined towards PBR radiance $L_G^{PBR}$, the updated surfel radiances are subsequently utilized to query the indirect illumination for the PBR radiance in the next iteration. As the surfel radiance adapts to the new lighting condition, the indirect lighting term in the PBR equation becomes more precise, which in turn provides a more accurate supervision signal back to the surfel radiances.
>
> The visual results in Figure 19 show that rendered images using the surfel radiance $L_G$ visually converge towards the PBR rendered image using  $L_G^{PBR}$ (second row) and get closer to the GT relight image. The bottom row, where the rendered $\mathcal{L}_{rad}$ starts with high error but swiftly fades to black, confirms that both PBR radiance $L_G^{PBR}$ and surfel radiance $L_G$ converge based on the new lighting condition.
>
> [1] Áfra, Attila T. "Open Image Denoise." 2019,
>
> [2] Hadadan, Saeed, Shuhong Chen, and Matthias Zwicker. "Neural radiosity." ACM Transactions on Graphics (TOG) 40.6 (2021): 1-11.
>
> [3] Su, Rui, et al. "Dynamic neural radiosity with multi-grid decomposition." SIGGRAPH Asia 2024 Conference Papers. 2024.
>
> [4] Coomans, Arno, et al. "Real‐time neural rendering of dynamic light fields." Computer Graphics Forum. Vol. 43. No. 2. 2024.

---

> ### Comment · Reviewer_47iG · 2025-11-24
> **Official Comment by Reviewer 47iG**
>
> I highly appreciate the efforts the authors have done, which make the paper more self-contained and easy to follow.
>
> The additional experiment results demonstrate the effectiveness of the proposed radiometric consistency loss. To be specific,
> 1. The additional visualization results of indirect lighting and estimated environment map shown that the proposed method achieve better indirect illumination modeling.
> 2. Fig.19 and the authors' response clarify the optimization dynamics during the finetuning process. And the fast convergence also shows that the proposed loss achieves the consistency between the surfel's outgoing radiance and the PBR result.
> 3. From my point view, the key insight of the method is that the surfel's outgoing radiance in unseen direction is misaligned with the PBR result, and this gap hinders the optimization process from converging to the ideal state. And the proposed method bridges the surfel's outgong radiance and the PBR result, effectively synchronizes this two components and mitigates the gap between them.
>
> Based on the reasons mentioned above, I will raise my score to "accept" and thank the author for their efforts during the rebuttal.

---

> > ### Author Response · Authors · 2025-11-25
> >
> > We are deeply grateful for the reviewer’s time and effort on providing thorough reviews and the feedbacks that improve our paper.

---

### Official Review · Reviewer_mzwa · 2025-10-31

**Soundness:** 2
**Presentation:** 3
**Contribution:** 2
**Rating:** 6
**Confidence:** 4

**Summary:**

This paper tackles the key problem of poor indirect illumination modeling in Gaussian Splatting-based inverse rendering. It proposes a physical constraint called radiometric consistency, designs an inverse rendering framework named RadioGS, and develops a fast finetuning-based relighting strategy. The radiometric consistency constraint provides supervision for unobserved viewpoints by minimizing the residual between the learned radiance of Gaussian surfels and their physically rendered radiance. This forms a self-correcting feedback loop. The RadioGS framework integrates 2D Gaussian ray tracing to deploy the constraint efficiently. The relighting strategy can adapt to new illumination conditions in minutes, with rendering latency below 10 ms. Experiments show the method outperforms existing Gaussian-based methods in novel view synthesis, geometry and material reconstruction, and relighting.

**Strengths:**

1. The paper is intuitive and easy to follow.
2. By combining the surfel radiance represented by spherical harmonics with physically based rendered radiance, the idea is straightforward and effective.
3. Through carefully designed loss functions, path tracing, and multi-stage training, the geometry reconstruction and inverse rendering have achieved superior results on multiple datasets.

**Weaknesses:**

1. The experimental validation is conducted exclusively at the object level, lacking evaluation on complex scene-level datasets. The scalability of the proposed framework to scene-level settings is a significant concern. The core Monte Carlo sampling strategy, which is computationally intensive even for objects, would likely incur a prohibitive overhead when applied to large-scale scenes with a massive number of Gaussian surfels.
2. The performance of the inverse rendering framework is dependent on the quality of a pre-trained novel view synthesis model, introducing a potential dependency and a point of failure.

**Questions:**

1. Scaling the method to scene-level settings presents primary challenges beyond the current object-centric experiments. Could the authors discuss the anticipated computational bottleneck in such a scenario? A detailed analysis of how the sampling parameters (e.g., N_g, N_s) and the associated rendering time are expected to scale would be crucial for assessing the method's practicality.

---

> ### Author Response · Authors · 2025-11-21
> **Response to Reviewer mzwa (1/2)**
>
> ### W1 & Q1. Scalability and Computational Cost on Scene-Level Settings
>
> We appreciate the reviewer’s concern about the scalability of our method. To provide further analysis of scalability, we deliver computational cost analysis of both the object-level TensoIR dataset and the scene-level MipNeRF360 dataset, varying the number of sampled Gaussian surfels per iteration, $N_g$, and the number of incident rays per surfel, $N_s$. All ablation studies are conducted on an NVIDIA RTX4090 GPU.
>
> **TensoIR dataset**
>
> Firstly, we evaluated the performance by varying $N_g$ from $1024$ to $8192$ while keeping $N_s$ fixed at 64. As shown in the table below, increasing $N_g$ generally improves reconstruction quality across all tasks and metrics by providing radiometric consistency supervision for more surfels per iteration. Also, $N_g$ does not impact the rendering cost, as it only controls the number of surfels supervised by radiometric consistency during the optimization step.
>
> | $N_g$ | NVS PSNR $\uparrow$ | NVS SSIM $\uparrow$ | NVS LPIPS $\downarrow$ | Normal MAE $\downarrow$ | Albedo PSNR $\uparrow$ | Albedo SSIM $\uparrow$ | Albedo LPIPS $\downarrow$ | Relight PSNR $\uparrow$ | Relight SSIM $\uparrow$ | Relight LPIPS $\downarrow$ | Render (ms) |
> | :---: | :---: | :---: | :---: | :---: | :---: | :---: | :---: | :---: | :---: | :---: | :---: |
> | 1024 | 37.8206 | 0.9799 | 0.0268 | 3.6900 | 30.9137 | 0.9518 | 0.0730 | 32.0569 | 0.9529 | 0.0481 | 38.51 |
> | 2048 (Ours) | 37.858 | 0.9801 | 0.0266 | 3.6889 | 31.0479 | 0.9521 | 0.0721 | 32.092 | 0.9533 | 0.0478 | 38.64 |
> | 4096 | 37.8707 | 0.9802 | 0.0264 | 3.6852 | 31.0495 | 0.9522 | 0.0721 | 32.1112 | 0.9532 | 0.0478 | 38.73 |
> | 8192 | 37.8799 | 0.9802 | 0.0263 | 3.6834 | 31.0507 | 0.9523 | 0.0720 | 32.1294 | 0.9533 | 0.0477 | 38.59 |
>
>
> We also analyzed the effect of $N_s$ ranging from $16$ to $128$ with $N_g$ fixed at $2048$. The table below demonstrates that reconstruction quality improves as $N_s$ increases up to $64$. However, increasing $N_s$ further to $128$ yields diminishing returns, and in some tasks (e.g., NVS, Relight), performance slightly drops, likely because the additional ray samples do not resolve the variance in Monte Carlo integration.
> Regarding efficiency, rendering cost scales with $N_s$, due to the additional computation for the Monte Carlo estimate for $L_G^{PBR}$. However, the rendering cost remains managably low even when using N_s=128, showing 58.3ms per frame (~17.2 fps)
>
> | $N_s$ | NVS PSNR $\uparrow$ | NVS SSIM $\uparrow$ | NVS LPIPS $\downarrow$ | Normal MAE $\downarrow$ | Albedo PSNR $\uparrow$ | Albedo SSIM $\uparrow$ | Albedo LPIPS $\downarrow$ | Relight PSNR $\uparrow$ | Relight SSIM $\uparrow$ | Relight LPIPS $\downarrow$ | Render (ms) |
> | :---: | :---: | :---: | :---: | :---: | :---: | :---: | :---: | :---: | :---: | :---: | :---: |
> | 16 | 37.4199 | 0.9792 | 0.0275 | 3.7174 | 30.8656 | 0.9505 | 0.0735 | 32.1684 | 0.9518 | 0.0494 | 16.23 |
> | 32 | 37.8125 | 0.9799 | 0.0268 | 3.6979 | 30.9014 | 0.9510 | 0.0730 | 32.1537 | 0.9528 | 0.0483 | 22.66 |
> | 64 (Ours) | 37.8707 | 0.9802 | 0.0264 | 3.6852 | 31.0479 | 0.9521 | 0.0721 | 32.1112 | 0.9532 | 0.0478 | 38.64 |
> | 128 | 37.6548 | 0.9798 | 0.0268 | 3.6821 | 30.9563 | 0.9512 | 0.0728 | 31.9768 | 0.9526 | 0.0480 | 58.30 |
>
> **MipNeRF360 dataset**
>
> We further provide a rendering cost analysis of the scenes in the MipNeRF360 dataset. Visual results of our method are in Figure 15 of the revised paper. The visual results were rendered using our model, trained with the same hyperparameters as in the paper.
>
> Overall, we compare the average rendering cost of our method applied to the TensoIR and MipNeRF360 datasets in the table below. As the reviewer noted, our method shows higher rendering costs than the results on the TensoIR dataset due to the increased number of Gaussian surfels. However, our method achieved rendering speeds below 100ms (>10 fps) on the MipNeRF360 dataset.
>
> | Method | Rendering (ms) |
> | :---: | :---: |
> | TensoIR | 38.64 |
> | MipNeRF360 | 81.40 |
>
> We measured the computational cost by varying $N_g$ from $1024$ to $16384$ while keeping $N_s$ fixed at 64, as presented in the table below. Similar to the analysis on the TensoIR dataset, $N_g$ brings a negligible impact on the rendering cost.
>
> | $N_g$ | Average (ms) | Bonsai (ms) | Counter (ms) | Kitchen (ms) | Room (ms) |
> | :---: | :---: | :---: | :---: | :---: | :---: |
> | 1024 | 81.23 | 77.37 | 72.38 | 86.81 | 88.43 |
> | 2048 (Ours) | 81.40 | 77.78 | 72.95 | 84.29 | 90.56 |
> | 4096 | 81.69 | 78.87 | 72.19 | 85.28 | 90.41 |
> | 8192 | 81.53 | 77.75 | 73.34 | 85.22 | 89.82 |
> | 16384 | 81.86 | 77.75 | 73.24 | 86.08 | 90.38 |

---

> ### Author Response · Authors · 2025-11-21
> **Response to Reviewer mzwa (2/2)**
>
> We also provide results on computational cost in the table below, varying $N_s$ from 16 to 64 while keeping N_g fixed at 2048. Increasing $N_s$ also increased rendering cost. However, our method could achieve faster rendering by using fewer ray samples per surfel.
>
> | $N_s$ | Average (ms) | Bonsai (ms) | Counter (ms) | Kitchen (ms) | Room (ms) |
> | :---: | :---: | :---: | :---: | :---: | :---: |
> | 16 | 24.57 | 23.49 | 23.86 | 26.07 | 24.85 |
> | 32 | 43.43 | 42.00 | 39.32 | 47.31 | 45.10 |
> | 64 (Ours) | 81.40 | 77.78 | 72.95 | 84.29 | 90.56 |
>
> In summary, the rendering cost of our method increases when applied to scene-level settings, but it can be controlled by adjusting $N_s$. Reducing $N_s$ to reduce rendering cost may come at the expense of rendering quality, but the analysis of TensoIR shows that increasing $N_g$ improves rendering quality. Therefore, our method may scale to larger scenes while maintaining computational efficiency by effectively manipulating both $N_g$ and $N_s$.
>
> __________________________________________________________________________________________________
> ### W2. Constrained Quality on Pre-trained Novel-view Synthesis Model
>
> We thank the reviewer for this important question regarding the dependency on pre-trained models, and apologize for the confusion caused by the phrasing in our original manuscript. We would like to clarify that our method does not rely on a standard NVS pre-training model. Instead, we employ a specialized "warm start" strategy that incorporates a simplified version of radiometric consistency via a split-sum approximation [1].
>
> A conventional NVS initialization is deployed to obtain reliable geometry for inverse rendering. However, we discovered that NVS initialized Gaussian surfels lead to sub-optimal performance. To this end, we replace the standard NVS pre-training by already incorporating our radiometric consistency as a simplified version using the split-sum approximation[1]. As detailed in Appendix A.5, the split-sum approximation efficiently estimates the integral of the rendering equation in Eq. (3) by decomposing the integral into diffuse and specular parts, both represented using preintegrated environment maps and lookup tables, without modeling indirect illumination.
>
> The table in Figure 6 of the submitted paper compares the performance of the model optimized with our initialization (Ours) and with the NVS initialization (NVS init.). The results show enhanced performance of our initialization stage compared to NVS pre-training.
>
> We further provide an additional visualization in Figure 16 of the revised paper, comparing our initialization against the NVS initialization for inverse rendering. As shown in the left-side plot, our initialization leads to substantially faster and more stable convergence of the radiometric consistency loss $\mathcal{L}_{rad}$, whereas the standard NVS initialization produces a higher and noisier residual throughout the training. This demonstrates that conventional NVS initialization places the surfels into a suboptimal orientation that is misaligned with the physical decomposition required for inverse rendering with radiometric consistency.
>
> The qualitative results on the right side of Figure 16 show that the NVS pre-training causes the model to misinterpret geometric cues, leading to incorrect albedo reconstruction (red arrows). Likewise, inter-reflections are often merged into the reconstructed albedo (blue arrows), indicating that indirect illumination cannot be reliably separated. In contrast, our initialization preserves physically priors from the beginning of optimization, enabling enhanced geometry reconstruction and cleaner separation of inter-reflection effects.
> In conclusion, our framework enforces physical constraint from the start of the optimization, ensuring robust Gaussian orientation for inverse rendering compared to novel-view synthesis pre-training.
>
> [1] Munkberg, Jacob, et al. "Extracting triangular 3d models, materials, and lighting from images." Proceedings of the IEEE/CVF Conference on Computer Vision and Pattern Recognition. 2022.

---

> > ### Comment · Reviewer_mzwa · 2025-11-27
> >
> > I appreciate the authors' detailed responses. The provided experimental results and visualizations addressed my concerns about the scalability and performance. Thus, I would like to raise my score to 8.

---

> > > ### Author Response · Authors · 2025-11-27
> > >
> > > Dear reviewer mzwa,
> > >
> > > We are deeply grateful for the reviewer’s time and effort on providing thorough reviews and the feedbacks that improve our paper.

---

### Official Review · Reviewer_MPqe · 2025-11-01

**Soundness:** 3
**Presentation:** 2
**Contribution:** 2
**Rating:** 2
**Confidence:** 4

**Summary:**

This paper tackles the problem of inverse rendering with 3D Gaussian Splatting, specifically focusing on the challenge of accurately modeling indirect illumination. Existing methods often struggle to disentangle material properties from global illumination effects because they lack supervision for indirect radiance from unobserved viewpoints. To this end, the paper proposes Radiometrically Consistent Gaussian Surfels (RadioGS), an inverse rendering framework that integrates radiometric consistency using 2D Gaussian ray tracing to efficiently compute indirect illumination and visibility.

**Strengths:**

The paper is well-written. The paper demonstrates state-of-the-art performance on two standard benchmarks (TensoIR, Synthetic4Relight). The quantitative tables and qualitative figures (especially the superior handling of inter-reflections in Figures 1 and 5) convincingly show the benefits of the proposed approach over existing methods. The proposed finetuning-based relighting method is very fast (<10ms), making the framework practical for applications requiring real-time rendering under dynamic illumination.

**Weaknesses:**

1. A primary concern is the conceptual framing of "radiometric consistency". The paper presents this as a "novel physical constraint", but its fundamental distinction from standard methods for computing indirect illumination, such as path tracing, is unclear. The underlying physical constraint is simply the rendering equation. This method appears to be a form of optimization where a learned or cached representation of global illumination is regularized to match a physically-based render. If this interpretation is correct, then radiometric consistency is not a new physical principle but rather an efficient approximation or caching strategy designed to enforce it. The novelty would then lie in the caching technique itself, not the underlying physics.

2. The formulation of the indirect radiance term, $L_\text{ind}$, is critically underspecified. Despite being a cornerstone of the proposed method, the paper lacks a clear mathematical definition for how $L_\text{ind}$ is computed. The text refers readers to Eq. 4 and Eq. 5 (line 240), but this reference is confusing and unhelpful. Eq. 4 defines the BRDF, and Eq. 5 merely decomposes incoming radiance into direct and indirect components without providing a computational model for the latter. The method's implementation for this crucial term is left to textual descriptions (e.g., lines 240, 268), which is insufficient for a clear understanding and reproduction of the work.

3. The evidence supporting the central claim of improved indirect illumination modeling is indirect and appears weak. The paper validates its method via ablation studies on end-to-end metrics (e.g., NVS, albedo PSNR) across a limited set of scenes. However, these final metrics can be influenced by many factors in the rendering pipeline. Without a direct, quantitative evaluation of the indirect illumination component itself—for instance, by comparing the rendered indirect pass against a ground-truth path-traced equivalent—it is difficult to conclude that the performance gains stem specifically from more accurate indirect radiance estimation. Relying on proxy metrics from a small number of scenes makes the current conclusion insufficiently substantiated.

**Questions:**

1. See W1. Is the radiometric consistency a form of optimization where a learned or cached representation of global illumination is regularized to match a physically-based render?
2. What is the codebase for implementation? Currently, it seems that the method is built upon IRGS's implementation.

---

> ### Author Response · Authors · 2025-11-21
> **Response to Reviewer MPqe (1/3)**
>
> ### W1 & Q1. Conceptual Framing of Radiometric Consistency
>
> We appreciate the reviewer’s comment regarding the framing of radiometric consistency. We agree with the reviewer’s assessment that our method is a form of optimization that guides Gaussian surfel radiances to cache global illumination by enforcing consistency with the physically-based rendered radiance. Based on the assessment, we would like to clarify about our radiometric consistency and how our framework contributes to Gaussian-Splatting (GS)-based inverse rendering.
>
> We would like to emphasize that our goal is not to introduce a new physical principle. As mentioned in line 053 of the submitted paper, our idea is motivated by self-training neural radiance caches[1,2]. In addition, as Reviewer 47iG noted in Strength #1, the key contribution of our work is in formulating radiometric consistency as a self-supervision signal that enables caching of radiances in unobserved directions within a GS-based inverse rendering framework. To further clarify the motivation behind our conceptual framing and why the proposed contribution is non-trivial, we demonstrate the key challenge in applying radiometric consistency to GS-based inverse rendering.
>
> Applying radiometric consistency to existing GS-based inverse rendering for enhanced indirect illumination is difficult because these pipelines cannot dynamically query surfel radiances as indirect radiances during optimization. As mentioned in lines 257~261 of the submitted paper, existing GS-based inverse rendering methods employ point-based ray tracing, which precomputes indirect radiance from NVS-pretrained Gaussian primitives in a non-differentiable manner. These precomputed values remain fixed during the optimization. When these values do not reflect updated Gaussian attributes, the supervision signal from radiometric consistency may become inconsistent.
>
> Our framework addresses this issue by employing differentiable Gaussian ray tracing to query indirect radiance for sampled surfels at every iteration. We further conducted ablation studies to show how our framework enhances the contribution of radiometric consistency in Gaussian-based inverse rendering. Starting from the model initialized with our method, we trained three ablation models using different methods for querying indirect radiances and rendering, while utilizing our radiometric consistency loss during inverse rendering optimization.
>
> - "Split-sum": Applies split-sum approximation to calculate physically-based radiance $L_G^{PBR}$ during the inverse rendering optimization, which does not involve indirect illumination.
> - "RT precompute": Applies Monte Carlo estimate (Eq.(8)) to calculate $L_G^{PBR}$ and precomputes indirect radiances via ray tracing, freezing the indirect illumination estimate during the optimization.
> - "RT w/o diff.": Dynamically updates indirect radiances via ray tracing during optimization, but without updating through ray-traced surfels by detaching the gradient on the ray-traced results.
>
> | Method         | NVS PSNR ↑ | NVS SSIM ↑ | NVS LPIPS ↓ | Normal MAE ↓ | Albedo PSNR ↑ | Albedo SSIM ↑ | Albedo LPIPS ↓ | Relight PSNR ↑ | Relight SSIM ↑ | Relight LPIPS ↓ |
> |----------------|------------|------------|--------------|---------------|----------------|----------------|-----------------|-----------------|-----------------|------------------|
> | Split-sum      | 36.659     | 0.9800     | **0.0254**   | 4.0066        | 27.471         | 0.9339         | 0.0828          | 28.816          | 0.9371          | 0.0575           |
> | RT precompute  | 35.547     | 0.9728     | 0.0332       | 3.7138        | 30.523         | 0.9481         | 0.0750          | 31.965          | 0.9516          | 0.0496           |
> | RT w/o diff    | 37.252     | 0.9789     | 0.0275       | 3.6946        | 30.673         | 0.9507         | 0.0732          | 32.050          | 0.9524          | 0.0484           |
> | Ours        | **37.858** | **0.9801** | 0.0266       | **3.6889**    | **31.048**     | **0.9523**     | **0.0719**      | **32.092**      | **0.9533**      |  **0.0478**       |

---

> ### Author Response · Authors · 2025-11-21
> **Response to Reviewer MPqe (2/3)**
>
> The results of the three ablation models and our method are depicted in the table above. "Split-sum" shows severe degradation in normal, albedo, and relighting accuracy due to the lack of illumination effects from surfels. "RT Precompute" shows enhanced normal, albedo, and relighting accuracy by utilizing precomputed indirect illumination, but yields the lowest novel-view synthesis performance among all methods. While "RT w/o Diff" improves overall performance through dynamic updates to these values, it still falls short of our method across all metrics. Ours method achieves the best performance across all metrics, demonstrating that the fully differentiable self-correcting feedback loop is essential for robust disentanglement.
>
> In summary, while radiometric consistency itself is not a new physical law, our contribution is to make this principle available in Gaussian surfels, allowing surfel radiances on unobserved views to be supervised based on radiometric consistency and to effectively utilize differentiable ray tracing to optimize surfel radiances as indirect radiances for inverse rendering.
>
>
> __________________________________________________________________________________________________
> ### W2. Formulation on Indirect Radiance Term and Gaussian Ray Tracing
>
> We thank the reviewer for pointing out the lack of clarity in the estimation of indirect radiance and visibility. We revised the corresponding lines in Sec 4.1.2 to properly describe the process by which we use Gaussian ray tracing to estimate indirect radiance and visibility, which we also provide below. Given a ray with origin x and direction $\omega_i$, the 2D Gaussian ray tracer, $\mathrm{Trace}\left(x,\omega_i\right)$ gathers
> $K$ Gaussian surfels intersecting the ray and returns both accumulated radiance $L_{trace}$ and the final transmittance $T_{trace}$ following the standard alpha-blending process from Eq.(2) as below:
>
> $$
> L_{\text{trace}} = \sum_{k=1}^K T_k \alpha_k L_k(x_k,-\omega_i), T_k = \prod_{m=1}^{k} (1 - \alpha_m),T_{trace} = T_K
> $$
>
> where $L_k(x_k,-\omega_i)$ denotes outgoing radiance towards the ray origin from the surfel $k$ with center $x_k$ and opacity $\alpha_k$. We then define indirect radiance $L_{ind}(x,\omega_i)$ and visibility $V(x,\omega_i)$ used in Eq.(5) as below:
>
> $$
> L_{ind}(x,\omega_i)=L_{\text{trace}}, V(x,\omega_i)=1-T_{trace}.
> $$
>
> We have also updated Appendix A.3 to include a detailed process for engineering details and a backward process for completeness. We hope our response and revision resolve the reproducibility concern.
>
> __________________________________________________________________________________________________
> ### W3. Evaluation for Indirect Illumination
>
> We thank the reviewer for the insightful feedback on the evaluation of indirect illumination. Current inverse rendering benchmarks, including TensoIR, Synthetic4Relight, and StanfordORB, do not provide ground truth (GT) for indirect illumination for quantitative and qualitative evaluation. To address the concern, we generated a new evaluation dataset with explicit GT Indirect Illumination.
>
> We utilized the original Blender files from the TensoIR dataset to generate high-fidelity ground truth. We used Blender Cycles path tracing engine to render the indirect illumination pass along with the original render pass. To ensure noise-free references, especially for indirect illumination, we set the sampling rate to 256 spp and applied the OIDN denoiser[3]. This allows direct quantitative evaluation of the indirect illumination components. We trained our model and the ablation model discarding the radiometric consistency loss (Ours w/o $\mathcal{L}_{rad}$) using the same hyperparameters described in the paper. We also trained two baselines, IRGS and SVG-IR, for comparison. We will make our dataset publicly available upon acceptance of the paper.
>
> The table below presents the quantitative comparison against baselines (IRGS, SVG-IR) and our ablation model on our new dataset. Our method significantly outperforms all baselines in indirect illumination reconstruction, confirming that our method accurately models the physical transport of indirect light. Our accurate indirect illumination leads to superior performance in most other metrics. When radiometric consistency is removed (Ours w/o $\mathcal{L}_{rad}$), the indirect PSNR drops significantly, showing that the performance gain comes from our proposed framework utilizing radiometric consistency, which effectively supervises indirect radiance from unobserved views.

---

> ### Author Response · Authors · 2025-11-21
> **Response to Reviewer MPqe (3/3)**
>
> | Method        | NVS PSNR ↑ | NVS SSIM ↑ | NVS LPIPS ↓ | Indirect PSNR ↑ | Indirect SSIM ↑ | Indirect LPIPS ↓ | Normal MAE ↓ | Albedo PSNR ↑ | Albedo SSIM ↑ | Albedo LPIPS ↓ |
> |---------------|------------|------------|--------------|------------------|------------------|-------------------|--------------|----------------|----------------|-----------------|
> | IRGS          | 35.0982    | 0.9660     | 0.0436       | 24.2219          | 0.8792           | 0.1092            | 4.1835       | 30.0931        | **0.9521**         | 0.0752          |
> | SVGIR         | 36.9634    | 0.9786     | 0.0266       | 30.9747          | 0.9134           | 0.0843            | 4.2624       | 29.7354        | 0.9309         | 0.0822          |
> | Ours w/o rad  | 36.3471    | 0.9764     | 0.0276       | 30.0954          | 0.9161           | 0.0752            | 3.8332       | 30.3425        | 0.9470         | 0.0760          |
> | Ours       | **37.8519** | **0.9822** | **0.0212**    | **32.8832**       | **0.9266**        | **0.0726**         | **3.6048**    | **30.6224**     | 0.9502       | **0.0744**        |
>
> We have also attached qualitative comparisons in Figure 13 of the revised paper. As shown, our method faithfully reconstructs indirect illumination compared to baseline models and our ablation model. Overall, we observed that our method produces indirect illumination closest to the ground truth, while IRGS produces overestimated intensity, and SVG-IR tends to underestimate the intensity of the indirect radiances. Certain phenomena can also be observed in the qualitative results in Figures 7 and 8 of our submitted paper. Our ablation model (Ours w/o $\mathcal{L}_{rad}$) tends to produce white blurs on inter-reflecting regions compared to our method due to the lack of supervision on unseen views. Finally, we provide additional qualitative comparisons of indirect illumination during relighting in Figure 14, where our method produces more realistic, accurate indirect illumination than the baseline models.
>
> __________________________________________________________________________________________________
> ### Q2. Codebase of our work
>
> Our implementation is based on the public codebase of IRGS, which we adopted for its efficient 2D Gaussian ray tracer. We introduced some modifications to support our framework.
>
> 1. We disabled the view-dependent normal-flipping mechanism in IRGS. This ensures that our surfel normals remain fixed and consistent, which is essential for the stable optimization of our radiometric consistency loss $\mathcal{L}_{rad}$.
> 2. To enable per-Gaussian PBR shading for inverse rendering and radiometric consistency, we have adopted the codebase from R3DG and SVG-IR. Note that IRGS uses Gaussian ray tracing to query indirect illumination for deferred shading.
>
> We will make our complete codebase publicly available upon acceptance.
>
>
> [1] Hadadan, Saeed, Shuhong Chen, and Matthias Zwicker. "Neural radiosity." ACM Transactions on Graphics (TOG) 40.6 (2021): 1-11.
>
> [2] Hadadan, Saeed, et al. "Inverse global illumination using a neural radiometric prior." ACM SIGGRAPH 2023 Conference Proceedings. 2023.
>
> [3] Áfra, Attila T. "Open Image Denoise." 2019.

---

> > ### Comment · Reviewer_MPqe · 2025-11-23
> >
> > I highly appreciate the authors’ detailed rebuttal and their effort in conducting extensive additional experiments.
> >
> > ---
> >
> > ### **W1**
> >
> > While I am convinced of the effectiveness of the proposed consistency term, I believe the current introduction or abstract still lacks precise statements. Specifically, the abstract states: “To address this issue, we introduce radiometric consistency, a physical constraint that provides supervision towards ….” I still believe it is better to say “introduce a novel physically based constraint” rather than “introduce a novel physical constraint”. This is because the proposed consistency is “a form of optimization that guides Gaussian surfel radiances to cache global illumination by enforcing consistency with the physically based rendered radiance”, as the authors agreed, or “a novel radiometric consistency loss” (Reviewer 47iG), or a novel approach “that provides principled physical supervision” (Reviewer ShdJ), rather than a new physical constraint, which could be misleading in my opinion.
> >
> > ---
> >
> > ### **W2**
> >
> > Thanks. It is clear to me now.
> >
> > ---
> >
> > ### **W3**
> >
> > While I don't think the rebuttal stage should be simply a matter of asking/conducting a large number of experiments, these experiments added by the authors do (in my opinion) crucially improve the paper's soundness. Specifically,
> >
> > In the initial paper, the validation of “more accurate indirect lighting” was not only qualitative but, more importantly, included only comparisons between methods without ground truth (Figs. 5/6/7). Without ground truth, unless one is a complete expert in indirect lighting (with qualitative intuition capable of estimating indirect lighting directly from RGB images), it is often impossible to distinguish the subtle differences between methods and determine which is better. This issue is common in many object-level inverse rendering works (I understand this is because common datasets such as Synthetic4Relight and TensoIR do not provide ground truth indirect lighting).
> >
> > An exception is MIRRES [1]. Although they also focus on multi-bounce path tracing, their ablation study (Fig. 7) is already convincing enough, and the method’s performance is strong (Fig. 6), so the lack of ground truth indirect lighting does not significantly undermine their persuasiveness.
> >
> > However, for the original submission of this work, the lack of ground truth is fatal because the core claim is to estimate more accurate indirect lighting using radiometric consistency. Figs. 6/8, used to support this claim, offer little compelling evidence (at least not enough to demonstrate significance), especially when the results fall far short of MIRRES (their Fig. 6).
> >
> > In this context, the authors provided ground truth indirect lighting during the rebuttal. The updated Figs. 13/14 more convincingly demonstrate that the significantly improved indirect lighting quality, compared to previous 3DGS-based methods. Furthermore, the authors have promised to release this as a dataset. Although preparing such a dataset is not particularly complex, doing so encourages the community to establish new evaluation standards, which I believe is valuable in itself. Therefore, I believe that this work has made a meaningful attempt to bridge the gap between 3DGS-based inverse rendering and mesh-based inverse rendering like [1].
> >
> > Considering these points, I am raising my rating to 6. However, in its current form, I still have two questions:
> >
> > ---
> >
> > ### **Q1**
> >
> > Similar to Reviewer 47iG’s W1, while I understand that the consistency loss is effective because the precomputed indirect radiance from NVS is not differentiable and thus not optimizable, I do not understand why another reason for its effectiveness lies in supervision towards unseen directions. Both Lines 195–201 and the response to Reviewer 47iG W1 are not clear to me. Could you please explain the right part of Fig. 2 in detail (especially the meaning of the arrows)? It seems that during NVS, the green Gaussian is unseen and can therefore learn arbitrary radiance to overfit the indirect term on the bottom gray Gaussian. But with consistency, the green Gaussian is constrained by “emitting = receiving,” so the indirect term will be constrained rather than arbitrarily learned as a residual to overfit RGB supervision. Is this interpretation correct?
> >
> > ---
> >
> > ### **Q2**
> >
> > I think one of the most important issues for 3DGS-based inverse rendering lies in its gap compared to mesh-based methods (e.g., MIRRES seems to offer a much more accurate disentanglement of direct/indirect lighting than any 3DGS-based methods). What is the key reason? For example, a recent work [2] suggests that the ill-posed nature of 3DGS geometry matters. I would appreciate it if the authors could elaborate on this.
> >
> > ---
> >
> > > *[1] Inverse Rendering using Multi-Bounce Path Tracing and Reservoir Sampling (ICLR'25)*
> > >
> > > *[2] GeoSplatting: Towards Geometry Guided Gaussian Splatting for Physically-based Inverse Rendering (ICCV'25)*

---

> ### Author Response · Authors · 2025-11-25
> **Additional Response to Reviewer MPqe (1/2)**
>
> We greatly appreciate the reviewer’s acknowledgement on our response, and for time and effort the reviewer dedicated to providing further constructive suggestions. Below, we provide responses towards additional suggestions and discussions.
>
> ### W1. Statement of "Radiometric Consistency"
>
> We thank the reviewer for pointing out the terms that causes confusion of our work’s novelty. Based on your suggestion, we have revised the term “physical” to “physically-based” throughout the paper.
>
> _________________________________________________________________________________________________
>
> ### Q1. Supervision Towards Unseen Direction
>
> We appreciate the reviewer for the opportunity to clarify our effectiveness in supervision towards an unseen direction, and also the details in Figure 2.
>
> In our paper, an “unseen” or “unobserved” direction refers to a direction that is not covered by the novel-view synthesis (NVS) training cameras. In typical inverse-rendering benchmarks, training viewpoints rotate around the object while looking towards its center, so NVS-pretrained Gaussian radiances are mainly optimized to synthesize images towards these limited camera directions. As a result, radiance queried along other directions (e.g., between nearby surfels for indirect illumination) is weakly or not at all constrained by NVS alone.
>
> The reviewer’s interpretation of Fig. 2 is correct. The figure shows a green Gaussian that is occluded from all training views, but whose outgoing radiance towards the bottom gray Gaussian still contributes to the gray surfel’s indirect term. The black-dotted arrows indicate how NVS supervision propagates via camera rays. Because of occlusion, the green Gaussian receives no direct NVS supervision, and even when differentiable Gaussian ray tracing is used at the gray surfel, there is no explicit constraint on the outgoing radiance of the green Gaussian along the direction toward the gray one. As the reviewer pointed out, this allows the green Gaussian to learn arbitrary radiance that overfits the gray surfel’s indirect term under NVS supervision.
>
> The pink-dotted arrows represent the supervision introduced by our radiometric consistency. We use incident radiances to compute a physically based radiance, and for a sampled outgoing direction (e.g., from the green to the gray Gaussian), we minimize the residual between the surfel radiance and this physically based radiance. In other words, our loss enforces the “emitting = receiving” constraint described by the reviewer by explicitly matching surfel radiance to physically rendered radiance, even along directions that are never seen by the cameras. We revised the lines 195~199 and the captions on Figure 2 of the revised paper.
>
> To further support our interpretation that the radiometric consistency supervises unseen directions, we conducted an additional ablation where we train both “Ours” and “Ours w/o $\mathcal{L}_{rad}$” on only 50% and 25% of the randomly subsampled training views of our new dataset. The numerical results on NVS and indirect illumination reconstruction performance are in the table below. We have denoted the performance drop relative to the full training view (100%) on the right side of the metrics. NVS metrics degrade for both methods when fewer views are used. However, when using 25% of the training views, indirect illumination reconstruction with our model remains nearly unchanged (-0.17dB), whereas the ablation shows a significant drop in indirect PSNR (-2.21dB). This indicates that when the camera viewpoint is limited, radiometric consistency still provides effective supervision that cannot be provided by NVS alone.
>
> | Train views | NVS PSNR $\uparrow$ (Ours) | NVS PSNR $\uparrow$ (Ours w/o $\mathcal{L}_{\text{rad}}$) | Indirect PSNR $\uparrow$ (Ours) | Indirect PSNR $\uparrow$ (Ours w/o $\mathcal{L}_{\text{rad}}$) |
> | :---: | :---: | :---: | :---: | :---: |
> | 100% | 37.85 | 36.35 | 32.88 | 30.10 |
> | 50% | 37.54 (-0.31) | 35.81 (-0.54) | 32.79 (-0.09) | 29.15 (-0.95) |
> | 25% | 36.79 (-1.06) | 34.65 (-1.70) | 32.71 (-0.17) | 27.89 (-2.21) |
>
> We hope our response and revision resolve the concern.

---

> ### Author Response · Authors · 2025-11-25
> **Additional Response to Reviewer MPqe (2/2)**
>
> ### Q2. Gap between GS-based Mesh-based Inverse Rendering
>
> We appreciate this insightful question regarding the gap between GS-based and mesh-based inverse rendering. We elaborate on the gap and position our work as follows.
>
> As the reviewer noted, the semi-transparent nature of Gaussian primitives makes it challenging to define surface intersections compared to explicit meshes. Since Gaussian primitives may not be perfectly opaque or aligned with the true surface, this geometric ambiguity can accumulate errors in visibility and shading, which may fundamentally limit the disentanglement quality in inverse rendering. To this end, MIRRES[1] utilizes a mesh representation extracted from the NeuS2[2], and GeoSplatting[3] exploits an optimizable mesh representation from FlexiCubes[4] for inverse rendering.
>
> However, modeling light transport in a physically accurate yet computationally efficient manner also remains challenging, even for mesh-based methods. MIRRES[1] leverages multi-bounce path tracing on the mesh to model indirect illumination, and mitigates Monte Carlo noise via reservoir sampling and image-space denoising. The authors of MIRRES explicitly argue for multi-bounce path tracing instead of querying indirect illumination from a neural radiance field (NeRF), since NeRF lacks physical constraints on indirect lighting. While MIRRES achieves enhanced disentanglement compared to GS-based methods, this comes with a relatively high computational cost of approximately 4.5 hours for training. Furthermore, to manage GPU memory efficiency, MIRRES does not backpropagate gradients through indirect rays. In contrast, our framework does not rely on heavy multi-bounce path tracing and noise-reduction methods. Instead, we use differentiable Gaussian ray tracing together with a radiometric-consistency loss so that ray-traced Gaussian surfels provide physically induced indirect radiance, and we explicitly propagate gradients through these indirect rays. This allows us to train efficiently within roughly 1 hour per scene while still enforcing physically grounded indirect illumination in the inverse rendering optimization.
>
> GeoSplatting[3] argues that 3DGS-based representations lack deterministic geometry boundaries and explicit normal definitions for inverse rendering. They address the limitation by aligning multiple opaque 3D Gaussians with optimizable normal vectors to each triangle of the mesh representation. Then they model indirect illumination through per-Gaussian residual shading terms for inverse rendering. In contrast, our method relies on disk-like Gaussian surfels (i.e., 2D Gaussians) with predefined normals as geometry representation and provides physically induced indirect illumination via radiometric consistency.
>
> We believe our framework is complementary and can serve as a building block that may be combined with such mesh-guided Gaussian representations. We have additionally cited GeoSplatting[3] in the related work section on the revised paper.
>
> [1] Dai, Yuxin, et al. "Inverse rendering using multi-bounce path tracing and reservoir sampling." arXiv preprint arXiv:2406.16360 (2024).
>
> [2] Wang, Yiming, et al. "Neus2: Fast learning of neural implicit surfaces for multi-view reconstruction." Proceedings of the IEEE/CVF International Conference on Computer Vision. 2023.
>
> [3] Ye, Kai, et al. "Geosplatting: Towards geometry guided gaussian splatting for physically-based inverse rendering." Proceedings of the IEEE/CVF International Conference on Computer Vision. 2025.
>
> [4] Shen, Tianchang, et al. "Flexible isosurface extraction for gradient-based mesh optimization." ACM Transactions on Graphics (TOG) 42.4 (2023): 1-16.

---

> > ### Comment · Reviewer_MPqe · 2025-11-27
> >
> > Dear ACs, Authors,
> >
> > I thank the authors for their detailed explanation. After reviewing the revised paper, I believe this work (in its current form) has been solid, insightful, and easily accessible. I really appreciate the key idea that the proposed consistency term can serve as an “emitting = receiving” constraint even along unobserved directions (inability of state-of-the-art mesh-based path tracing approaches), of which the significance I overlooked during my initial review. Also, during rebuttal, the additional experiments on indirect lighting evaluation have significantly enhanced the work's validity, which I find highly valuable. For these reasons, I am revising my initial rating (from an initial 2) to 8. I am willing to provide any in-depth justification for my rating if the ACs feel it necessary. Thanks to the authors' and the ACs' efforts and time.

---

> > > ### Author Response · Authors · 2025-11-28
> > >
> > > Dear Reviewer MPqe,
> > >
> > > We sincerely appreciate your time, careful reconsideration, and the thoughtful feedback that greatly strengthened our work. We are truly grateful for recognizing our contributions.

---

### Author Response · Authors · 2025-12-03
**Summary of Contribution and Responses**

Dear AC and reviewers,

We sincerely appreciate the reviewers for their constructive comments and suggestions. We provide a summary of our work’s contribution and the key outcomes of the discussion.  Revisions are colored blue in the paper.

# Contribution

Our work introduces a novel Gaussian Splatting (GS)-based inverse rendering framework with physically induced indirect illumination based on the three core contributions:

1. **Radiometric Consistency**, a novel physically-based constraint enabling Gaussian surfels to learn physically grounded radiance along unobserved directions by minimizing the residual between learned and physically rendered radiance [R-mzwa, R-ShdJ]
2. **RadioGS**, an inverse rendering framework that efficiently integrates radiometric consistency by deploying differentiable Gaussian ray tracing [R-mzwa, R-47iG]
3. **Finetuning-based Relighting** that quickly adapts surfel radiances to new lighting by enforcing radiometric consistency, achieving high rendering efficiency (\< 10ms per frame) [R-MPqe, R-47iG, R-ShdJ]

Our method shows **superior performance** on novel-view synthesis (NVS), geometry and material reconstruction, and relighting on two benchmarks. [R-MPqe, R-mzwa, R-47iG, R-ShdJ]

# Discussion

We have strengthened our work based on the discussion as follows:

| Discussions | R-MPqe | R-mzwa | R-47iG | R-ShdJ | Our Response |
| :---- | :---- | :---- | :---- | :---- | :---- |
| **Novelty on Radiometric Consistency** | ✔ |  |  | ✔ | 1. / Appx.F |
| **Indirect Illumination Evaluation** | ✔ |  | ✔ |  | 2. / Appx.E |
| **Scalability to scene-level settings** |  | ✔ |  |  | 3. / Appx.G |
| **Initialization Stage** |  | ✔ | ✔ |  | 4. / Appx.B\&H |
| **Finetuning-based Relighting** |  |  | ✔ | ✔ | 5. / Appx.J |
| **Environment Map** |  |  | ✔ |  | 6. / Appx.I |
| **Comparison with Mesh-based Methods** | ✔ |  |  |  | 7. / [comment](https://openreview.net/forum?id=lKqE7UuMvp&noteId=iEvPhFJUxY) |
| \--- | \--- | \--- | \--- | \--- | \--- |
| **Initial Scores** | 2 | 6 | 6 | 6 | \- |
| **Response Acknowledged** | ✔ | ✔ | ✔ | \- | \- |

1) We emphasized that existing pipelines lack a mechanism to enforce radiometric consistency, and **our contribution is the novel formulation that effectively integrates this principle via differentiable Gaussian ray tracing**. We provided ablation studies demonstrating how naive implementations fail and how our design successfully provides supervision for unobserved directions **(Tab.5&6)**. Reviewer MPqe acknowledged this distinction and agreed the issue was resolved.
2) Since current benchmarks lack ground-truth (GT) for indirect illumination, **we extended the TensoIR dataset by generating GT indirect illumination**. Results on our dataset show that **our method is quantitatively and qualitatively superior to baselines (Tab.4, Fig.13&14)**. We committed to publicize this dataset upon acceptance. The reviewers agreed the results are convincing, while reviewer MPqe noted that our dataset could help the community to establish new evaluation standards.
3) We conducted ablation studies on two sampling parameters in both object- and scene-level settings **(Tab.7-10, Fig.15)** to show **scalability of our work while maintaining rendering efficiency**. Reviewer mzwa responded that the issues are addressed.
4) We provided illustration and ablation studies **(Fig.16, Tab.3&11)** to **clarify the contribution of our initialization**. Reviewers confirmed that our response addressed the issue.
5) We provided demonstrations on the optimization dynamics, computation and memory cost analysis , and visual comparisons on relighting with ablation models **(Fig.19&20, Tab.12)** to show **the efficiency and quality of our method**. Reviewer 47iG noted that our response clarified the finetuning process
6) We provide qualitative comparisons on environment map reconstruction, which show **our method provides results that are visually closest to the GT environment map (Fig.17&18)**. Reviewer 47iG acknowledged our method’s illumination modeling performance.
7) We discussed the gap between GS-based and mesh-based inverse rendering methods. Reviewer MPqe noted that our work can address the inability of mesh-based methods in modeling illumination.

We further revised the paper for clarification as follows:

1. Refine terms from “physical” to “physically-based” \[R-MPqe\]
2. Details of supervision on unseen directions \[R-MPqe\] (Sec.4.1, Fig.2)
3. Details on Gaussian ray tracing and indirect radiance queries \[R-MPqe\] (Sec.4.1.2, Appx.A.3)

# Closing Remarks

After the discussion, reviewers MPqe, mzwa, and 47iG expressed that their comments and issues are fully satisfied and provided positive assessment. While reviewer ShdJ provided no further comments, the initial review highlights the novelty and performance as strengths. We hope that the discussions and reviewers' positive consensus will be considered in the final decision.

Best regards,
Authors

---

### Meta-Review · Area_Chair_oP6U · 2025-12-31

**Summary:**

# Decision

This submission contributes to 3DGS-based inverse rendering and relighting, combining surfel-based radiance represented with spherical harmonics and physically based rendering in a coherent and effective framework. Reviewers acknowledged the relevance of the problem, the soundness of the technical approach, and the convincing empirical results, with SOTA performance demonstrated on multiple benchmarks.

Crucially, the authors’ rebuttal was highly responsive and substantially improved the paper. Most initial concerns regarding methodological clarity, conceptual framing, and experimental validation were addressed to the reviewers’ satisfaction. Overall, the strength of the technical contribution, the clarity of the presentation, and the thorough and effective rebuttal support a clear accept recommendation.

------------
# Consolidated Initial Reviews

## Strengths

### Relevant technical contributions
- Effective contribution: combining the surfel radiance represented by spherical harmonics with physically-based rendered radiance [`mzwa`, ].
- Effective contribution: efficient finetuning-based relighting method [`MPqe`, `47iG`, `ShdJ`].
- Novel inverse-rendering framework [`47iG`].

### Convincing results
- SOTA performance on 2 standard benchmarks, with convincing quantitative + qualitative results [`MPqe`, `mzwa`, `47iG`, `ShdJ`].

### Misc.
- Well-written [`MPqe`, `mzwa`].

## Weaknesses

### Under-specified methodological aspects
- Questionable conceptual framing of the proposed radiometric consistency [`MPqe`].
- Limited novelty c.f. relatively straightforward [`ShdJ`].
- Under-specified formulation of the key radiance term $L_{\text{ind}}$ [`MPqe`].
- Unclear how $L_G$ can supervise unseen directions [`47iG`].

### Questions on applicability
- Method's dependency on the quality of a pretrained NVS model [`mzwa`].
- Extra finetuning required for relighting, limiting its applicability [`ShdJ`].

### Limited ablation study
- Weak support for the central claim of improved indirect illumination modeling [`MPqe`].
- Validation exclusively at the object level, not scene-level -- can the method scale to large-scale scenes (c.f. MC sampling overhead)? [`mzwa`]
- Unclear ablation study protocol w.r.t. initialization methods [`47iG`].
- Lack of ablation study w.r.t. impact of finetuning [`ShdJ`].

**Reviewer Concerns:**

See above for summary of main concerns initially shared by reviewers.

The authors addressed nearly all the above during rebuttal. Methodological aspects were clarified in detail, and further experimental results were provided, e.g., with additional ablations and explanations.

**Reviewer Scores:**

### Reviewer `MPqe`
- **Original score:** 2
- **Score change:** increased to 6 then 8, c.f. the reviewer's own satisfied responses to the authors' rebuttal.

### Reviewer `mzwa`
- **Original score:** 6
- **Score change:** increased to 8, c.f. the reviewer's own satisfied responses to the authors' rebuttal.

### Reviewer `47iG`
- **Original score:** 6
- **Score change:** increased to 8, c.f. the reviewer's own satisfied responses to the authors' rebuttal.

### Reviewer `ShdJ`
- **Original score:** 6
- **Score change:** likely to have kept their score, c.f. unresponsive (but the reviewer had their concerns decently covered by the authors).

---

### Decision · Program_Chairs · 2026-01-26

Accept (Oral)